# Advanced multi-modal mass spectrometry imaging reveals functional differences of placental villous compartments at microscale resolution

The placenta is a complex and heterogeneous organ that links the mother and fetus, playing a crucial role in nourishing and protecting the fetus throughout pregnancy. Integrative spatial multi-omics approaches can provide a systems-level understanding of molecular changes underlying the mechanisms leading to the histological variations of the placenta during healthy pregnancy and pregnancy complications. Herein, we advance our metabolome-informed proteome imaging (MIPI) workflow to include lipidomic imaging, while also expanding the molecular coverage of metabolomic imaging by incorporating on-tissue chemical derivatization (OTCD). The improved MIPI workflow advances biomedical investigations by leveraging state-of-the-art molecular imaging technologies. Lipidome imaging identifies molecular differences between two morphologically distinct compartments of a placental villous functional unit, syncytiotrophoblast (STB) and villous core. Next, our advanced metabolome imaging maps villous functional units with enriched metabolomic activities related to steroid and lipid metabolism, outlining distinct molecular distributions across morphologically different villous compartments. Complementary proteome imaging on these villous functional units reveals a plethora of fatty acid- and steroid-related enzymes uniquely distributed in STB and villous core compartments. Integration across our advanced MIPI imaging modalities enables the reconstruction of active biological pathways of molecular synthesis and maternal-fetal signaling across morphologically distinct placental villous compartments with micrometer-scale resolution.

The placenta is an organ that has diverse roles critical for establishing and maintaining pregnancy, including anchoring the fetus to the uterine wall[1], functioning as an immune barrier between mother and fetus[2], and regulating maternal metabolism to ensure a supply of nutrients to the fetus, thereby supporting fetal growth and development[3]. The hemomonochorial human placenta has a cotyledonary structure in which the villous trees function as the site of nutrient and oxygen uptake[4,5]. As depicted in Fig. 1, the villous functional unit in the term placenta is comprised of syncytiotrophoblast (STB), a multinucleated cellular structure formed by the fusion of trophoblastic cells into a syncytium, which covers the entire surface of the villous tree and provides the primary barrier between maternal

✉ e-mail: lisa.bramer@pnnl.gov; myattl@ohsu.edu; Kristin.Burnum-Johnson@pnnl.gov

**Fig. 1 | Placenta histology.** Syncytiotrophoblast (STB) - outer villous compartment that along with discontinuous cytotrophoblast cover the entire surface of the villous tree. STB provides a barrier between maternal blood and fetal blood. Core – the inner villous compartment that consists of fetal blood vessels, mesenchymal stromal cells, macrophages and connective tissue. pFEC placental – fetal endothelial cell.

blood on the outside and fetal blood within the villous core. In addition to serving as a physical barrier, the STB layer is also a critical site for maternal-fetal exchange and the synthesis of steroid and peptide hormones that regulate maternal metabolism and fetal growth and development[6]. The fetal blood vessels are embedded in a non-cellular matrix (connective tissue) along with mesenchymal stromal cells and fetal macrophages (Hofbauer cells), forming the villous core[7]. Thus, the human placenta is a cellularly heterogeneous tissue[8], characterized by significant molecular and histological changes throughout the progression of pregnancy[9]. Examining molecular variations and changes within the placental tissue microenvironment is crucial for understanding cellular heterogeneity and functional differences that are associated with adverse pregnancy outcomes.

"Omics" techniques are widely used in biomedical science[10,11] to explore the molecular mechanisms within different tissue types, thereby improving our understanding and prediction of the functions within complex biological systems. Placental structure and function are suspected to be closely related to long-term maternal and child health[12]. Therefore, investigating both morphological and molecular changes can enhance our understanding of the processes and mechanisms during pregnancy that contribute to the health of both the mother and the fetus. Previous omics studies have deciphered expression profiles of different omics levels[10] by profiling metabolites[13], lipids[14,15], proteins[16,17], transcrips[18–20]. However, these omics measurements profiled molecules from bulk samples, averaging biological processes and signaling across placenta tissue, obscuring crucial information on spatial localization. To overcome this challenge, single-cell and spatial omics technologies have been applied in biomedical science, to reveal the cellular heterogeneity of complex tissues and characterize poorly described subpopulations[21]. Several studies have applied single-cell transcriptomics to identify varying activities[22,23], including molecular divergence[24] and putative interactions[25], of individual placental cells. Single-nucleus multi-omic profiling of human placental STB provided researchers insights into the heterogeneity of STB, allowing them to identify cellular trajectories during pregnancy[26]. Additionally, spatially resolved single-cell multiomic characterization allowed scientists to describe the complete trophoblast invasion trajectory in early pregnancy[27].

An alternative method for examining cellular diversity and tissue heterogeneity is mass spectrometry imaging (MSI). MSI is a powerful technique that has been used for untargeted, in situ mapping of proteins, lipids, metabolites, and xenobiotics across biological tissue sections[28–30]. Previous studies utilized MSI with two commonly used ionization techniques, nanospray desorption electrospray ionization (nano-DESI) and matrix-assisted laser desorption/ionization (MALDI), to spatially resolve molecules at embryo implantation sites and placenta tissue. Coupling nano-DESI ionization to a high-resolution mass analyzer enabled imaging experiments with 10 µm spatial resolution to profile lipid and metabolomic compounds from mouse uterine sections[31]. Previously, MALDI-MSI has been used to characterize the spatial and temporal distribution of phospholipid species associated with mouse embryo implantation[32] and visualize alterations in the spatial distribution of phospholipids between terminal and stem villi in normal, uncomplicated, human-term placenta[33] and those with pathohistological maternal malperfusion[34]. With respect to comprehensive proteome imaging, a few ex-situ MALDI-MS approaches were reported, where proteins from placental tissue homogenate were first separated using one-dimensional or two-dimensional electrophoretic systems and then identified using MALDI-MS[35–37]. MSI does have certain limitations when it comes to in situ protein analysis. In the case of MALDI-MSI and (nano)DESI-MSI, both techniques were designed in a way that desorption of the analyte cannot be decoupled from subsequent ionization, therefore it's impossible to incorporate a separation component before ionization, limiting the dynamic range of observed analyte concentrations and restricting detection to the most abundant species. Although protein coverage can be improved with on-tissue digestion, confident in situ MS/MS peptide identification remains challenging due to low signal-to-noise ratios, ionization suppression, and high spectral complexity that impede database identifications[38]. Recent advances in single-cell and spatial proteomic technologies, such as bottom-up proteomic processing using droplet or nanowell-based sample preparation techniques, and enhancement of MS detection sensitivity, allow comprehensive proteome profiling by liquid chromatography-tandem mass spectrometry (LC-MS/MS) from samples with a limited tissue volumes down to single cells. As such, coupling Laser Capture Microdissection (LCM) cell isolation and

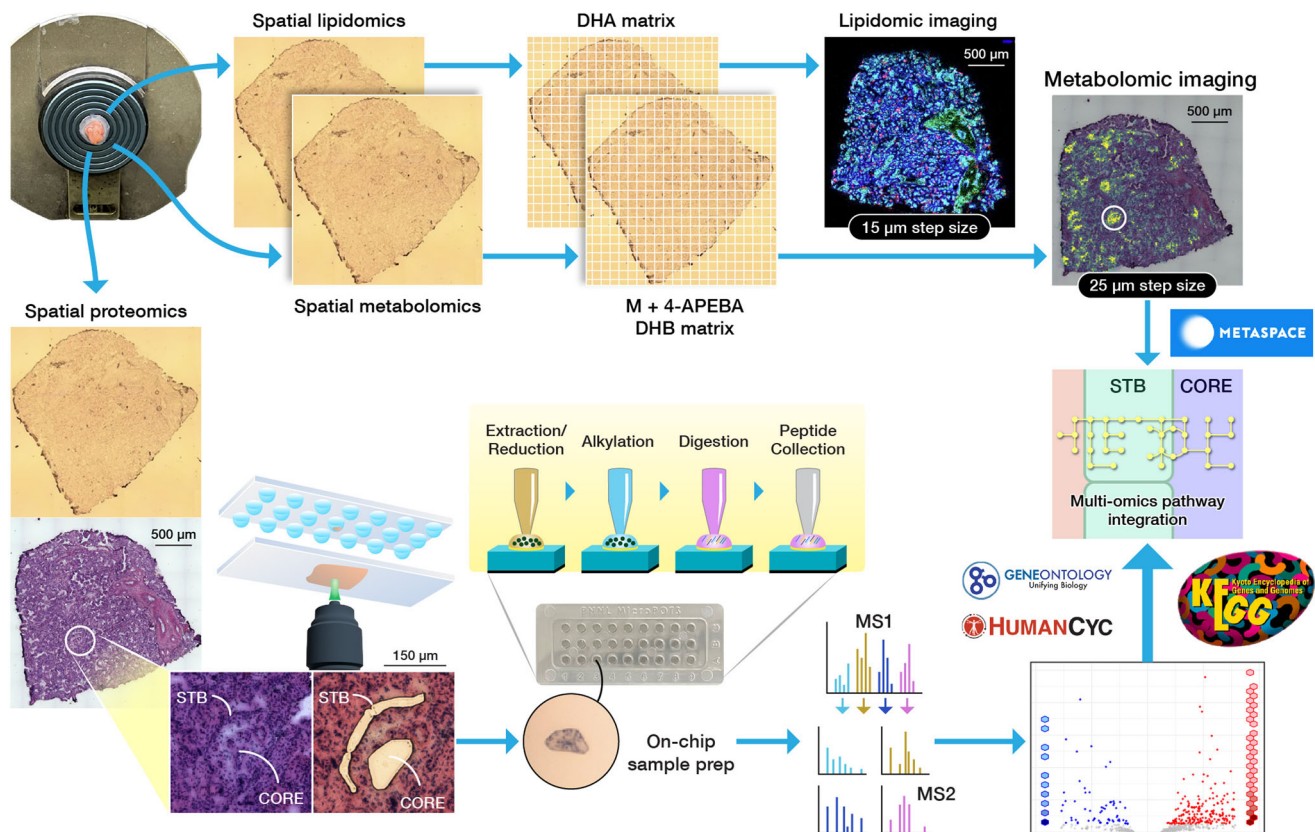

**Fig. 2 | Advanced MIPI approach.** Schematic workflow of the advanced MIPI approach that combines multi-modal MALDI-MSI for comprehensive lipidome and metabolome imaging with complementary microscale proteome profiling by microPOTS. 4-APEBA 4-(2-((4-bromophenethyl)dimethylammonium)ethoxy) benzenaminium dibromide, DHA 2,5-dihydroxyacetophenone, DHB 2,5-dihydroxybenzoic acid.

nanodroplet Processing in One pot for Trace Samples (nanoPOTS) processing, allowed researchers to perform cell-type-specific imaging across mouse uterine tissue sections preparing for embryo implantation, and resulted in >2000 identified proteins from 100 μm by 100 μm tissue pixels[38].

Integrating data from multiple omics levels enables a comprehensive understanding of molecular interactions driving complex biological processes[10,39]. The placenta is a complex tissue, exhibiting substantial substructural changes during normal development. Therefore, spatial multi-omics approaches are essential to capture heterogeneous changes in molecular dynamics and distribution across the tissue. The integration of MS-based techniques across multiple omics layers has been previously applied to the placenta, albeit with the limitation of using datasets derived from bulk samples[40,41]. A previous study integrated MS-based spatial proteomics data, with co-registered spatial transcriptomics profiles obtained by Digital Spatial Profiling (DSP), constructing a spatiotemporal atlas of the human maternal–fetal interface in the first half of pregnancy[42].

Although various MS-based workflows have been developed for spatial multi-omics integration[43–47], our previously published workflow, MIPI (Metabolome Informed Proteome Imaging), delivers pathway-level resolution across complex and heterogeneous biological systems[48]. The MIPI workflow utilizes MSI approaches to functionally reduce the heterogeneity of complex biological systems by spatially resolving microscale regions where specific activities are enriched, visualizing molecular composition and dynamic activities across 12-micron thick tissue sections. First, MALDI-MSI resolves the in situ molecular complexity with micrometer resolution by employing metabolomic measurements to pinpoint regions where specific activities are enriched. Next, the mapped regions are excised from adjacent

sections, for ultrasensitive proteomic analyses using microdroplet Processing in One pot for Trace Samples (microPOTS) to identify region-specific enzymes. Finally, the metabolomic and proteomic data are integrated and rolled up to the pathway-level to reconstruct underlying biological pathways within the micrometer-scale regions.

In this study, we advance our MIPI workflow[48] to broaden its utility in biomedical research. Although the successful application of MSI in profiling metabolites and lipids across placenta tissue has been demonstrated, the detection of endogenous steroid hormones by MSI remains challenging. The difficulty arises because steroid hormones lack hydrogen donor or acceptor moieties, which are necessary to facilitate ionization for MS analysis[49]. Therefore, we tailored our MIPI workflow to obtain the information needed to characterize lipid and steroid molecular mechanisms across and throughout human placenta tissue. As depicted in Fig. 2, we evolve our previously published MIPI workflow by incorporating lipidomic imaging to capture specific lipid signatures across placenta tissue sections, obtaining more comprehensive imaging data. Additionally, implementing on-tissue chemical derivatization (OTCD) into our metabolomic imaging workflow enhances sensitivity and enables us to detect endogenous steroid hormones and fatty acids and their metabolic dynamics. Multi-modal MALDI-MSI provides comprehensive molecular coverage of different omics-levels, allowing for precise mapping of micron-scale subregions of interest (sROIs) with enhanced metabolic activity. The integration of histological hematoxylin and eosin (H&E) staining in our MIPI workflow enhances the visualization of morphology and correlates it with molecular changes. Our complementary proteome profiling of mapped sROIs allows us to capture subregion-specific enzymes. Integration of the data from our multimodal MS-based approaches

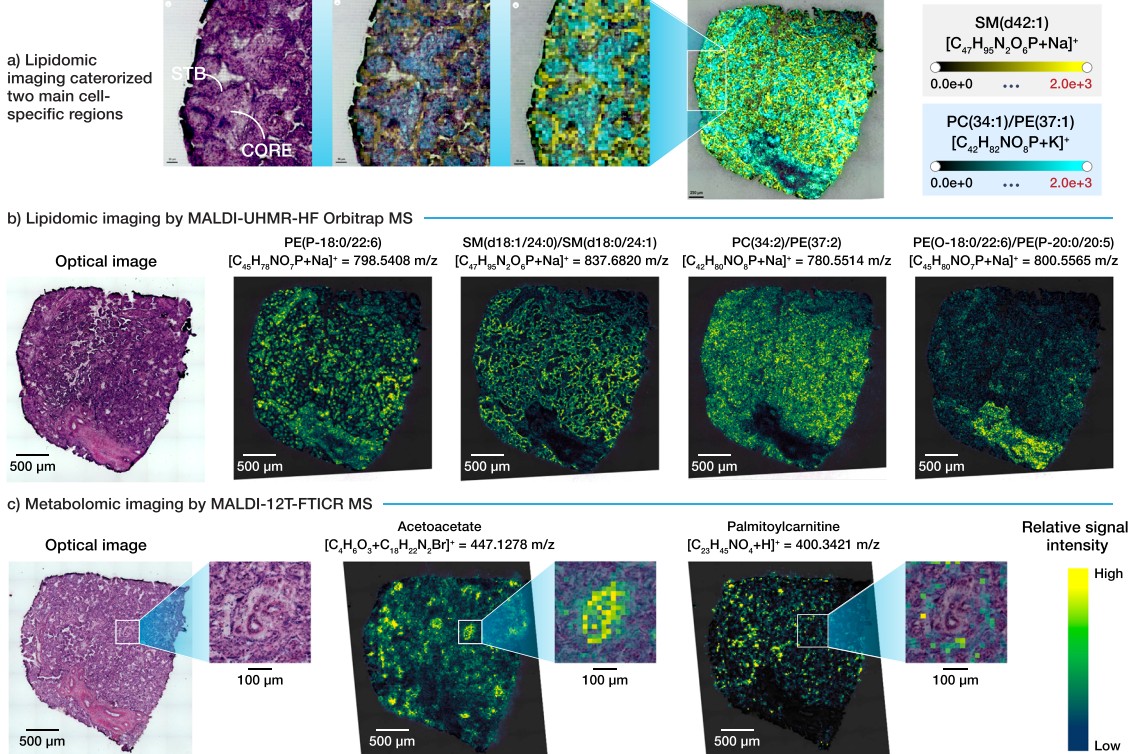

**Fig. 3 | Multi-modal MALDI MSI. a** MALDI-MSI categorized two main subregions, STB and core. **b** Lipidomic imaging by MALDI-UHMR-HF Orbitrap MS, performed on two adjacent sections. H&E-stained optical image and example ion images of section 1, from left to right in the following order: lipids localized in core and stem villi; lipid localized in STB; lipids present in both, STB and core, compartments; lipids detected in stem villi. Lipidomic imaging of placental section 2 is in Supplementary fig. 1. **c** Metabolomic imaging by MALDI-12T-FTICR MS, performed on two adjacent sections. H&E-stained optical image of placental section 3 and example ion images of acetoacetate and palmitoylcarnitine detected in core and STB, respectively. Metabolomic imaging of placental section 4 is in Supplementary fig. 1.

reconstructed underlying biological pathways relevant to placental function across histologically distinct villi compartments.

## Results

### Untargeted, multi-modal MALDI−MSI for comprehensive lipidomic and metabolomic profiling at a micrometer-scale level

Lipidomic imaging was performed on two adjacent 12 μm-thick placenta sections to resolve lipid signatures with 15 μm spatial resolution. By leveraging the METASPACE annotation platform to search against the LipidMaps database, we tentatively annotated 182 and 227 molecular features at a 20% false discovery rate (FDR) from two adjacent sections, respectively (Supplementary Data 1 and 2). This approach identified unique spatial patterns of lipid species across the placenta sections. Distinct molecular distributions of different lipids identified regional differences between the outer surface of the villi, comprised of the STB layer, and the villous core (Figs. 1 & 3a). As depicted in Fig. 3b, sphingolipids such as sphingomyelins and ceramide 1-phosphates were mainly detected in STB, as opposed to 1-(1Z-alkenyl),2-acylglycerophosphoethanolamines that were mainly detected in the villous core. Differential expression patterns of different diacylglycerophosphates, diacylglycerophosphoethanolamines, and diacylglycerophosphocholines were observed between the villous STB and core, with many of the lipids also being present in both compartments (Supplementary Data 3). Although lipid molecules detected in the villous stem region (Fig. 1) were also present in the villous core, some molecular signatures were exclusively detected in stem villi (i.e., Fig. 3b, tentative annotation PE(O-18:0/22:6)/PE(P-20:0/20:5)). Although stem villi exhibited some metabolomic differences compared to other imaged villi, these were not our primary focus. This is because the main function of stem villi is to provide structural support for villous trees, with

negligible involvement in fetal-maternal exchange and endocrine activity[50]. Next, comprehensive metabolomic imaging, incorporating the OTCD technique, was performed to identify hotspots of metabolic activity related to sterol lipids and fatty acid processing. This method was applied to two adjacent 12 μm-thick placenta sections to resolve metabolite signatures with 25 μm spatial resolution. By leveraging the METASPACE annotation platform to search against the CoreMetabolome database, we tentatively annotated 716 and 618 ion images from two adjacent sections, respectively, at 20% FDR (Supplementary Data 4 and 5). This enabled us to map the presence of a diverse array of metabolites, fatty acids, and lipid-related mediators with distinct spatial patterns. Our advanced metabolomic imaging workflow allowed us to visualize villous regions with enhanced metabolomic activities, highlighting subregional differences between the STB and the core compartments (Fig. 3c). The core compartment of mapped villous regions revealed the presence of metabolites tentatively annotated as acetoacetate and hydroxybutyric acid, suggesting an enhanced ketone body metabolism in this area. In contrast, the corresponding STB compartment exhibited the presence of steroids and various acylcarnitines, indicating high levels of steroid and fatty acid processing activities, respectively.

### Proteome profiling of metabolome-informed placental villous subregions

In contrast to lipidomic imaging, which showed comparable molecular abundance across all imaged villous functional units, advanced metabolomic imaging provided discerning molecular visualization, allowing us to find villous functional units with high metabolic activity related to fatty acid and steroid metabolism. Informed by the metabolome-specific villous regions (i.e., acetoacetate and

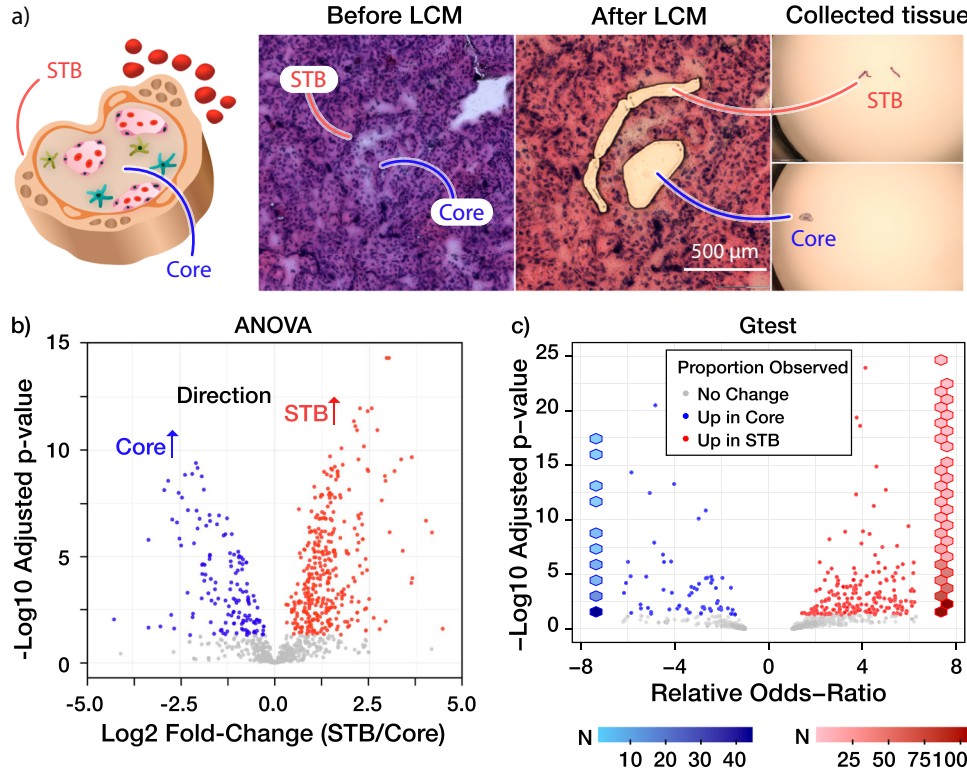

**Fig. 4 | Unveiling subregion-specific enzymes using microPOTS processing.**
**a** LCM collection of placenta villous subregions. **b** Volcano plot for the abundance-based model comparing the tissue subregion means for each protein. A mixed effects linear model, as fully described in the Methods section, with a conditional Gaussian distribution was fit to the normalized log relative abundance data for each protein. A two-sided likelihood ratio test for lack-of-fit was conducted and a Benjamini-Hochberg multiple comparison adjustment was calculated on all p-values. Source data are provided as a Source Data file. **c** Volcano plot for the probability of detection-based model comparing the tissue subregion mean detection probabilities for each protein. A generalized mixed effects linear model,

as fully described in the Methods section, with a conditional binomial distribution was fit to the binary detect/not detected outcome data for each protein. A two-sided likelihood ratio test for lack-of-fit was conducted and a Benjamini-Hochberg multiple comparison adjustment was calculated on all p-values. Source data are provided as a Source Data file. Red hexagons and blue hexagons represent proteins identified exclusively in the STB and Core, respectively. The color of each hexagon corresponds to the scale bars below the graph, which indicate the number of proteins (N). In the present volcano plot, extreme values were not shown. The plot with results for all proteins can be found in Supplementary Statistical Methods.

palmitoylcarnitine) (Supplementary Fig. 3), we complemented our metabolome data by profiling the proteomes of 14 villous regions (14 STB and 13 core subregions) from three adjacent sections (Supplementary Data 6–8). We collected and processed each villous STB and core sROIs separately (Fig. 4a). From a 10,000 μm² villous subregion pixel, we identified more than 12,600 unique peptides that mapped to more than 2800 unique proteins (Supplementary Data 6–8). A total of 439 and 185 proteins were found to be differentially expressed in the STB and core sROIs, respectively (ANOVA-adjusted *p*-values < 0.05, Supplementary Statistical Methods, Supplementary Data 9 and Fig. 4b). We found that the abundance of 769 and 177 proteins significantly increased in STB and core, respectively (G test adjusted p-value < 0.05, Supplementary Statistical Methods, Supplementary Data 9 and Fig. 4c). Over 20% of the detected proteins were differentially expressed in the two villous compartments, highlighting their distinct molecular compositions and suggesting subregion-specific metabolic pathways.

Enrichment analysis of differentially expressed proteins enabled us to associate these proteins with relevant biological processes and metabolomic pathways, functionally characterizing each villous compartment and revealing subregional differences related to metabolic processing (Supplementary Data 10). As shown in Supplementary fig. 4a, Gene Ontology (GO) enrichment revealed that STB subregions are enriched in biological processes related to the initiation and regulation of protein synthesis (cytoplasmic translational initiation,

positive and negative regulation of RNA splicing), as well as the aggregation and arrangement of molecules needed for the structural organization of DNA (nucleosome assembly) and protein production (formation of the cytoplasmic translation initiation complex, ribosome assembly, and U2-type prespliceosome assembly). The upregulation of protein biosynthesis in the STB is expected, as proper peptide hormone synthesis in the STB layer is crucial throughout gestation to support fetal development and sustain pregnancy[51]. A polypeptide hormone with peptide sequences corresponding to chorionic somatomammotropin hormones 1 and 2 (CSH1 and CSH2), also known as placental lactogen, which is synthesized during pregnancy, was significantly increased in the STB subregion (adjusted p-value of 2.18E-12). Lactogen is important for mother and fetus energy homeostasis[51], and can increase levels of free fatty acids (FFAs) through its lipolytic effects[52]. Pathway enrichment analysis using the Kyoto Encyclopedia of Genes and Genomes (KEGG) and the Encyclopedia of Human Genes and Metabolism (HumanCyc) databases revealed that the STB had a high prevalence of enzymes associated with the fatty acid beta-oxidation pathway, TCA cycle, hormone biosynthesis, and protein N-glycosylation (Supplementary Fig. 4b and c). The detection of several subunits of the oligosaccharyl transferase complex (RPN1, RPN2, STT3A, STT3B, DAD1, and DDOST with adj. p-values of 3.58E−49, 1.24E−34, 8.04E-15, 6.74E-10, 2.50E-8, and 3.60E-10, respectively), which is responsible for the initial transfer of a Glc₃Man₉GlcNAc₂ glycan from the lipid carrier dolichol-pyrophosphate to an

asparagine residue, catalyzes the first step in protein N-glycosylation. We also detected alpha-1,3-mannosyl-glycoprotein 2-beta-N-acetylglucosaminyltransferase (MGAT1, adj. *p*-value = 0.04), an enzyme involved in the synthesis of complex N-glycans. Profiling enzymes involved in N-glycosylation within the specific villous compartment demonstrated that our microscale profiling offers unique insights into region-specific protein glycosylation. This is important, as previous histochemical studies have revealed that placental structures with morphological and functional abnormalities display modified carbohydrate profiles[53]. Future efforts in analyzing placenta tissue using the workflow for N-glycan imaging[54,55] will provide information on the exact structures of the post-translational modifications. We also identified proteins of the annexin A family (ANXA1, ANXA4, ANXA5 and ANXA6 with adj. p-values of 6.02E-47, 2.27E-20, 6.99E-20, and 8.93E-13, respectively) as being prevalently expressed in STB. A previous study reported altered expression of annexins in women with reproductive disorders[56], indicating that detected proteins from the annexin family could be used as biomarkers for rapid diagnosis or therapeutic purposes. In contrast, the placental villous core, being a connective tissue layer that supports the fetal capillaries and villous trophoblast[57], was predictably enriched in processes related to extracellular matrix organization (including basement membrane organization, collagen fibril organization and keratinization) and regulation (positive regulation of integrin-mediated signaling pathway). For instance, the core was enriched with pericellular proteoglycans such as perlecan (HSPG2, adj. *p*-value = 3.21E−21), collagen alpha-1(XVIII) chain (COL18A1, adj. *p*-value = 2.77E−10) and agrin (AGRN, adj. *p*-value = 0.045). Additionally, small leucine-rich proteoglycan, decorin (DCN, adj. *p*-value = 1.18E-12) and biglycan (BGN, adj. *p*-value = 0.04) were enriched in the core. These proteoglycans play important roles in the structural and functional integrity of the placenta and fetal membranes[58]. Significant increases in macrophage mannose receptor 1 (MRC1, adj. *p*-value = 5.23E−10) and monocyte differentiation antigen (CD14, adj. *p*-value = 2.43E−5) indicate that the core is enriched in immunoregulatory proteins. Regarding the overexpressed metabolic pathways in the core, the upregulation of enzymes associated with energy metabolism was anticipated, given that late in gestation, fetal glucose metabolism is essential to the development of skeletal muscles, the fetal liver, the fetal heart, and adipose tissue[59].

## Spatial multi-omics integration provides a comprehensive overview of the active biological pathways at microscale resolution

Pathway enrichment analysis provided insights into significantly overrepresented pathways in each of the villous subregions, allowing us to map specific metabolic conversions and reconstruct pathways through manual integration of our multi-omics data. With a focus on elucidating metabolic pathways related to fatty acids and steroid processing, we aimed to reconstruct those processes. As depicted in Fig. 5a-i and Supplementary fig. 5a–d, the integration of multi-omics data generated through the advanced MIPI approach allowed us to transition from the molecular level to the pathway level, enabling the reconstruction of spatially resolved pathways that visualize differences in biological functions of the STB and core subregions.

The incorporation of OTCD into our metabolome imaging workflow enabled us to image progesterone, an endogenous steroid hormone, in the STB subregion (Fig. 5a). Complementary microscale proteomics profiling revealed the unique presence of 3 beta-hydroxysteroid dehydrogenase isomerase type 1 (HSD3B1, adj. *p*-value = 3.99E−36) and type 2 (HSD3B2, adj. *p*-value = 1.09E-32) enzymes in STB. These 3β-HSD type 1 and 2 enzymes convert pregnenolone to progesterone, and their colocalization with progesterone highlighted the estrogen synthesis pathway in the STB (Supplementary Fig. 5c). Our proteomics data revealed the upregulation of a diverse set

of enzymes in STB, effectively enabling the reconstruction of essential steps in the estrogen synthesis pathway[7] (Fig. 5b). The detection of mitochondrial cholesterol side-chain cleavage enzyme (CYP11A1, adj. *p*-value = 2.25E−41) reconstructed the first step of steroidogenesis, which involves the conversion of cholesterol to pregnenolone. The aromatase enzyme (CYP19A1), responsible for converting androgens into estrogens, was also identified in our dataset (adj. *p*-value = 4.29E−37). 17-beta-hydroxysteroid dehydrogenase type 1 (HSD17B1, adj. *p*-value = 1.04E-9), which converts estrone to the most potent estrogen (estradiol), was detected exclusively in STB. Conversely, 17-beta-hydroxysteroid dehydrogenase 2 (HSD17B2, adj. *p*-value = 8.85E−7), which catalyzes the oxidation of estradiol to the biologically less active estrone, was uniquely detected in the core. This suggests a unique, spatiotemporally specific hormone signaling pathway that protects the fetus from highly potent estradiol levels during the maternal-fetal exchange. Likewise, the fetus is also protected from excessive glucocorticoids by 11-beta-hydroxysteroid dehydrogenase type 2 enzyme (HSD11B2, adj. *p*-value = 1.51E−38), which was detected exclusively in STB. This enzyme metabolizes maternal cortisol into its inactive form, cortisone.

Detection of serine palmitoyltransferase 1 (SPTLC1, adj. *p*-value = 0.04) and 3-ketodihydrosphingosine reductase (KDSR, adj. *p*-value = 0.0003) suggested de novo synthesis pathway of ceramide in STB[60,61] (Fig. 5d). Mapped enzymes related to ceramide biosynthesis showed a high correlation with our lipidome imaging data, which captured ceramide-1-phosphates and sphingomyelins localized prevalently in STB (Fig. 5c). Although this pathway was not enriched in the STB due to the low number of significant proteins, we curated conversion of de novo synthesis pathway of ceramide as part of sphingolipid metabolism in STB, from the literature[60] and our lipidomic imaging data (Supplementary Fig. 5d).

The implementation of OTCD into our metabolomic imaging workflow allowed us to increase the selectivity and sensitivity of metabolites and detect endogenous FFAs as well. We discovered that chemically derivatized FFAs did not entirely colocalize with their esterified carnitine forms as depicted in Fig. 5e. For instance, oleoyl-carnitine and linoleyl carnitine were localized in STB subregions, while the corresponding derivatized oleic and linoleic acids were present across the entire villous functional unit, supporting the concept of FFAs transfer across the placenta. Proteomics data revealed the presence of fatty acid transport protein 2 (SLC27A2, adj. *p*-value = 0.04) in STB, suggesting that FFAs are transferred from maternal to fetal circulation, traversing the STB. Fatty acid binding proteins 4 and 5 (FABP4 and FABP5), detected in both villous compartments, can suggest that intracellular FFAs are bound and subsequently processed to membrane phospholipids, used for energy production in mitochondria, or esterified to triacylglycerols as the intermediate storage form in cells[62]. While spatial proximity doesn't always imply functional relevance, detection of various acylcarnitines (molecules tentatively annotated as palmitoylcarnitine, oleoylcarnitine, linoleyl carnitine, and stearoyl carnitine) by MALDI-MSI, suggests that FFAs localized in the STB layer may be shuttled toward energy production. Our complementary proteome profiling reconstructed the mitochondrial fatty acid β-oxidation pathway in STB[63] as depicted in Fig. 5f and Supplementary fig. 5a, by mapping the main enzymes of the pathway: (1) mitochondrial carnitine O-palmitoyltransferase 2 (CPT2, *p*-value = 0.04), responsible for transfer of long-chain fatty acid into the mitochondria; (2) a very long-chain specific acyl-CoA dehydrogenase (ACADVL, adj. *p*-value = 4.76E−20) that catalyzes the initial step of mitochondrial β-oxidation; (3) trifunctional enzyme subunits alpha and beta (HADHA and HADHB, adj. *p*-values of 2.72E-17 and 6.15E-6, respectively), which are involved in the last three steps of mitochondrial beta-oxidation of long chain fatty acids. We also mapped mitochondrial delta(3,5)-delta(2,4)-dienoyl-CoA isomerase (ECH1, adj. *p*-value = 2.03E−5), an auxiliary enzyme involved in the beta-oxidation

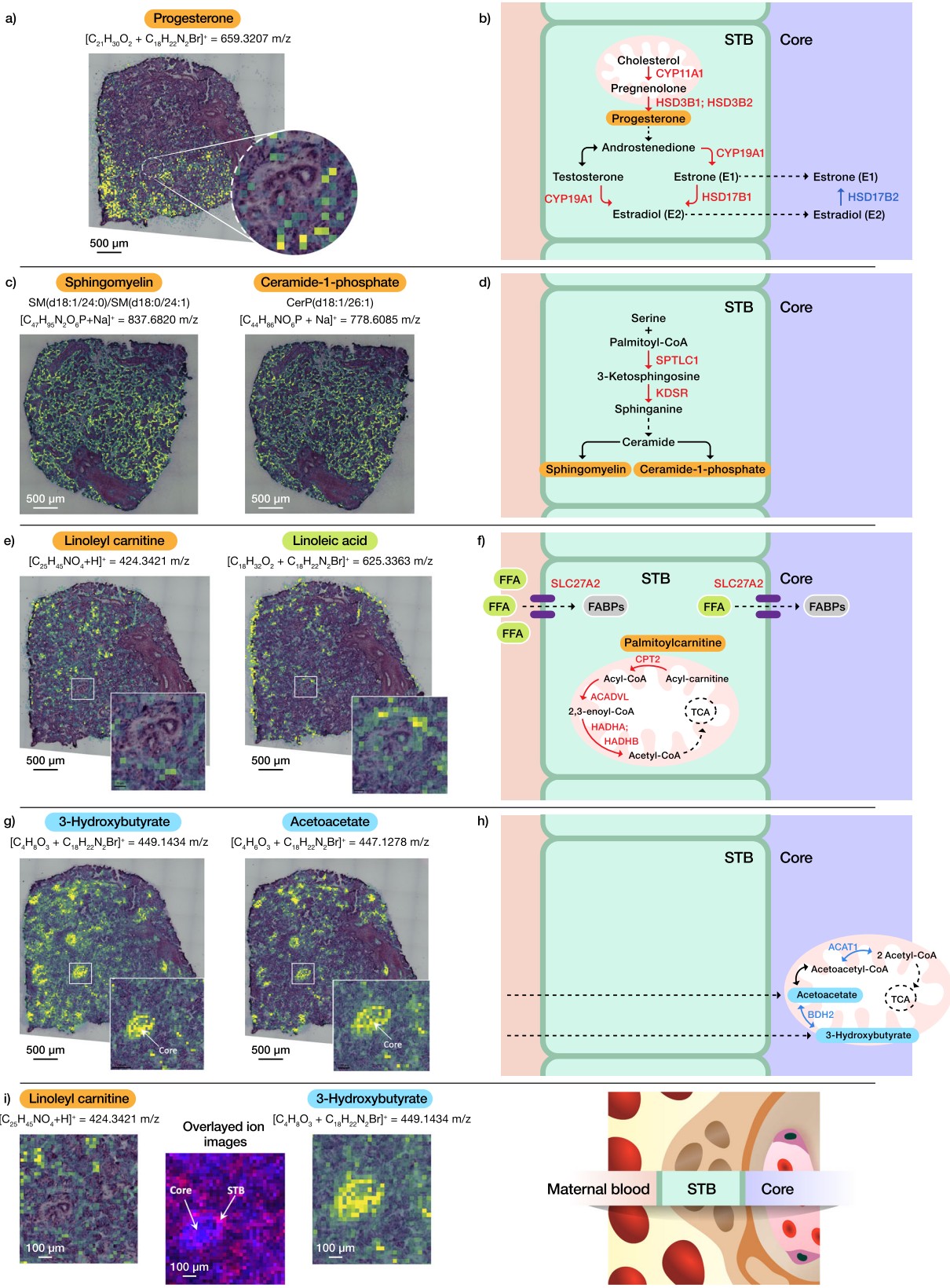

of unsaturated fatty acids. Several enzymes of the oxidative phosphorylation pathway were also detected, such as cytochrome c oxidase subunits, 5A, 6C, and 4 isoform 1 (COX5A, COX6C and COX4I1 with adj. $p$-values of 7.94E−6, 3.21E−5, 0.0001, respectively) in the mitochondrial electron transport chain. Additionally, the detection of lysophospholipid acyltransferase 5 (LPCAT3, adj. $p$-value = 4.31E-05) suggested

that intracellular FFAs can be alternatively remodeled in the Lands cycle and transported to fetus[63].

One of the pathways significantly overrepresented in the core was ketolysis. Our metabolome imaging showed the presence of the two main ketone bodies, 3-hydroxybutyrate and acetoacetate, exclusively detected in the core compartments (Fig. 5g–i). Complementary

**Fig. 5 | Spatial multi-omics integration unravels unique biological pathways in morphologically distinct placenta villi compartments. a** Advanced metabolomic imaging captured chemically derivatized progesterone in STB. Metabolomic imaging was performed on two adjacent sections. **b** Reconstructed estrogen synthesis and signaling pathways. Microscale proteomics was performed on three adjacent sections on 14 STB and 13 core subregions. **c** Lipidomic imaging revealed lipid signatures in STB. Lipidomic imaging was performed on two adjacent sections. **d** Reconstructed de novo synthesis pathway of ceramide in STB. Microscale proteomics was performed on three adjacent sections on 14 STB and 13 core subregions. **e** Linoleyl carnitine imaged in STB and corresponding linoleic fatty acid in both STB and core, indicating fatty acid transport across villi functional units. Metabolomic imaging was performed on two adjacent sections. **f** Reconstructed

model of fatty acid transport and energy production pathway. Microscale proteomics was performed on three adjacent sections on 14 STB and 13 core subregions. **g** Ketone bodies prevalently detected in the villous core. Metabolomic imaging was performed on two adjacent sections. **h** Proposed ketone body oxidation pathway in the villous core. Microscale proteomics was performed on three adjacent sections on 14 STB and 13 core subregions. **i** Overlayed ion images of acylcarnitine and ketone body, demonstrating compartmentalization of metabolic pathways in placental villi. Metabolomic imaging was performed on two adjacent sections. Enzymes in red color – detected in the STB; Enzymes in blue color – detected in the core; Proteins highlighted in grey were detected in both compartments. Molecules highlighted in orange, blue, and green were detected by MALDI-MSI with localization in STB, core, and both compartments, respectively.

proteome profiling mapped the mitochondrial dehydrogenase/reductase enzyme SDR family member 6 (BDH2, adj. $p$-value = 0.01) that may convert 3-hydroxybutyrate to acetoacetate, reconstructing ketone body metabolism (Fig. 5h and Supplementary Fig. 5b). Suggested ketone body conversion showed high substrate–enzyme–product spatial correlation in the core subregions. Ketone bodies are likely further converted to acetoacetyl-CoA, which is ultimately oxidized to two acetyl-CoA molecules by mitochondrial enzyme acetyl-CoA acetyltransferase (ACAT1, adj. $p$-value = 0.001), to form usable energy in the core.

## Discussion

In this paper, we advanced our previously developed MIPI workflow and demonstrated its application in bio-medical science to elucidate molecular processes at microscale resolution. To prove MIPI's ability to functionally reduce the heterogeneity and complexity of human tissue, we utilized a full-term placenta from an obese person collected post cesarean delivery. While the placenta regulates both maternal metabolism and fetal growth and development, it remains one of the least studied organs[64]. Previous publications of MS-based measurements on placenta tissue have mainly leveraged single-omics techniques, which only reflect a portion of the biochemical processes related to lipids, metabolites or enzymes. Although those omics approaches had some spatial context by sampling different sub-anatomical regions (maternal, middle, and fetal side), processing and analyzing bulk samples masks the detection of the low-abundance and highly localized pathways that might be of interest. Therefore, to demonstrate the importance of investigating molecular dynamics within the full spatial context of the microenvironment, we utilized our MIPI approach to capture biological processes and maternal-fetal signaling across placenta tissue.

Knowing that the placenta is one of the principal tissues in steroid hormone production[65], we were particularly interested in elucidating molecular processes related to steroid metabolism and signaling pathways. Imaging steroids with conventional MALDI-MSI workflows is challenging due to their low ionization efficiency. Therefore, we enhanced the detection sensitivity of steroids in our MIPI workflow by incorporating OTCD into MALDI-MSI metabolomics imaging. For OTCD, we used a combined application of EDC and 4-APEBA, which was previously demonstrated to surpass the benefits of conventional derivatization agents used for carboxylic acid and aldehyde/ketone derivatization[66]. Our advanced workflow ensured that the FFAs detected and discussed throughout this manuscript are endogenous and not fragments of lipids artificially generated by the MALDI laser. OTCD also allowed us to capture progesterone in its derivatized form, which was exclusively localized only in the STB compartment. Progesterone imaging, along with complementary proteome profiling, enabled us to reconstruct the estrogen synthesis pathway in the STB, culminating in the final conversion of estrone to the most potent form of estrogen (estradiol) by HSD17B1. Estrone and estradiol may also be formed by the action of steroid sulfatase (STS) on estrone sulfate and estradiol sulfate, respectively, which was also detected in our proteomics data (adj. $p$-value = 2.31E−28). Our advanced metabolomic

imaging captured a conjugated metabolite of estrone, estrone glucuronide, localized in both villous compartments, indicating that estrogens are likely stored and transported in their conjugated form. Detection of HSD17B2 in the core indicates a conversion pathway that operates in the opposite direction, leading to estrone accumulation. This aligns with previous findings that HSD17B2 is localized in endothelial cells of fetal blood vessels within the core, acting as a barrier and protecting the fetus by decreasing estradiol secretion rates into the fetal circulation[67].

We further enhanced our MIPI workflow by incorporating lipidomic imaging. Complementary lipidomic and metabolomic imaging techniques expand detection coverage and deliver more comprehensive data, which can precisely guide subsequent proteome profiling of microscale regions. Thus, our versatile MIPI workflow enables the interrogation of metabolome- and lipidome-guided regions, ranging from tissue functional units to subunits. For this specific application, we enhanced the spatial resolution of our MALDI-MSI measurements by using 25 μm and 15 μm step sizes for metabolomic and lipidomic imaging, respectively, compared to the previously used 50 μm spatial resolution, which was better suited for imaging sparse biological material[48]. By leveraging our microdroplet processing platform, we reliably analyzed subregional differences in protein levels within villous compartments. Consequently, we demonstrated improved sensitivity in our proteomics imaging by collecting and processing smaller tissue amounts (10,000 μm² tissue area) compared to our previous MIPI experiments, which profiled regions ranging from 400,000 to 700,000 μm² in area. This demonstrates the versatility of the MIPI approach, allowing it to be tailored to specific application needs by focusing on lipidome- and metabolome-guided imaging with complementary proteome profiling, and selecting the appropriate spatial resolution. We envision that MIPI will continue to evolve, ultimately overcoming the current limitation in exploring biomolecular signatures in tissues at the cellular level. For example, a sublimation method for matrix deposition can produce an even layer of crystals with a diameter of less than 1 μm, which, combined with highly focused lasers, can approach single-cell resolution for lipid and metabolite imaging. Leveraging nanodroplet processing for spatial proteomics can enable pixel-to-pixel matching and complementary profiling at a single-cell level.

Although MALDI-MSI can employ analyzers with ultra-high-mass resolving power and mass accuracy, confident molecular annotation remains a challenge for this stand-alone method due to isomerism. Bulk data is often used to complement MALDI-MSI data; however, it lacks spatial information, the molecular signals are usually diluted, and molecular profiling is not fully comprehensive due to the use of a different ionization technique (electrospray ionization). Confident molecular identification with full spatial context of the tissue can be achieved by combining MALDI-MSI with trapped ion mobility spectrometry (TIMS). As such, tentatively annotated molecules from untargeted MALDI-MSI can be confirmed with a targeted imaging approach by MALDI-TIMS-MS, along with standards confirmation. Alternatively, integration of multi-omics data can provide a higher degree of certainty in the molecular annotation of lipids and

metabolites by colocalizing the detection of their corresponding enzymes. Mapping enzymes and their substrates and products across multiple sections can suggest functional relationships, providing pathway-level insights at micrometer-scale resolution. Although co-detection does not imply co-regulation nor distinguish between pathway crosstalk, it provides spatial context that can be used to propose potential pathways, aiding in understanding basic biology and disease processes in specific tissue types. With this study, we demon-strated the capability of integrated MIPI data to provide a compre-hensive view of underlying pathway activities within different villous compartments. This capability can enable more effective investiga-tions of maternal-fetal interactions and enhance our understanding of placental function across different gestational stages and maternal conditions. For instance, our integrated MIPI data mapped the ketone body oxidation pathway in the villous core by detecting colocalized substrate, enzyme, and product. Among the various molecules in the STB layer, our proteomics data revealed the presence of lactogen hormone, which is synthesized by the STB[68]. As previously reported, lactogen promotes lipolysis in human adipose tissue in vitro, increas-ing the circulating FFA, which can be used as an energy source by producing ketone bodies[51]. Mapping the ketone bodies in the villous core aligns with previous knowledge that ketone bodies can freely pass across the placenta and be used as substrate for lipid and cholesterol synthesis[69], or fuel for oxidative metabolism by the fetus[70], thus vali-dating our reconstructed ketone body oxidation pathway in the villi core compartment. This example hypothesis clearly demonstrates the power of integrative spatial multi-omics analyses to provide detailed, pathway-level insights not obtainable with bulk measurements or sin-gle imaging approaches.

Here, we demonstrated MIPI's versatility and capability to tailor omics-specific workflows for obtaining precise molecular-level information, mapping elusive molecular dynamics across human placenta tissue sections. Future MIPI experiments will aim to incorporate another omics modality by leveraging spatial tran-scriptomics to investigate expressed RNA species. Integrating spa-tial transcriptomics into MIPI can enable even more comprehensive molecular profiling, providing insights into gene expression in specific cells at a given time. This can further augment our under-standing of the placental transcriptome in development and pathology. The integration and correlation of expressed genes with active biological pathways can yield even greater biological and clinical significance. A current limitation of our MIPI workflow is the requirement for multiple adjacent tissue sections, as different modalities necessitate the use of specific sample slides. Therefore, future efforts will focus on enabling MIPI to perform multi-omics profiling from a single tissue section while retaining or further improving the sensitivity of all modalities.

## Methods
### Materials
Polypropylene microwell chips with 2.2 mm wells diameter were manufactured on polypropylene substrates by Protolabs (Maple Plain, MN). Ethanol was purchased from Decon Labs, Inc (King of Prussia, PA). LCMS grade water, formic acid (FA), methanol (MeOH), acetoni-trile (ACN), triethylamonium bicarbonate (TEAB), iodoacetamide (IAA), and dithiothreitol (DTT) were purchased from Thermo Fisher Scientific (Waltham, MA). N-Dodecyl β-d-maltose (DDM), DMSO (HPLC grade), Phosphate-Buffered Saline (PBS), 1-ethyl-3-(3-(dimethylamino) propyl)carbodiimide (EDC), 2,5-dihydroxyacetophenone (DHA), 2,5-dihydroxybenzoic acid (DHB) were purchased from Sigma-Aldrich (St. Louis, MO). Lys-C and trypsin were purchased from Promega (Madi-son, WI). 4-(2-((4-bromophenethyl)dimethylammonium)ethoxy) ben-zenaminium dibromide (4-APEBA) was synthesized by a previously described procedure[66]. Hematoxylin and Eosin (H&E) stain kit was purchased from Abcam.

### Tissue collection
Informed consent was obtained from a patient under a protocol approved by the Institutional Review Board of OHSU (IRB #16328), prior to the tissue being de-identified and placed in a tissue repository. Our research complied with all of the relevant ethical regulations and followed and was performed according to the Declaration of Helsinki. Placental tissue was collected post cesarean delivery at term (39 weeks) and belonged to a female fetus. Inclusion criteria included a singleton pregnancy, an age range of 18–45 and delivery by cesarean section with no labor. Villous tissue (VT) was immediately sampled from a random site of the placenta, flash frozen and stored at −80 °C.

### Cryosectioning
Fresh frozen placenta tissue sample was mounted on a cryomicrotome chuck by freezing a small droplet of water and then cut into 12-μm-thick sections using a CryoStar NX70 (Thermo Fisher) with a blade temperature of −20 °C and specimen temperature of −18 °C. Adjacent placenta sections were collected on the indium tin oxide (ITO) slides for MALDI-MSI ($n = 4$) and on polyethylene naphthalate (PEN) mem-brane slides ($n = 3$) for subsequent laser-capture microdissection ana-lysis. Sections on PEN slides were washed with a gradient of ethanol solutions (70%, 96% and 100% ethanol, respectively) for 30 s during each change to dehydrate the sections. The collected sections with their cryosectioning order in parentheses were as follows: lipidomic sections 1 (section 8), lipidomic section 2 (section 28), metabolomic section 3 (section 18), metabolomic section 4 (section 22), and 3 adjacent proteomics sections (sections 19, 23 and 26).

### Metabolomic and lipidomic imaging by MALDI-MSI
Metabolomic imaging was completed by spraying an aqueous solution of EDC at 6 mg/mL first with subsequent application of 4-APEBA at 2 mg/mL using an external syringe pump with the M5 Sprayer (HTX Technologies, Chapel Hill, NC) using methods described in greater depth elsewhere[66,71]. Spraying parameters were the same for both chemicals: a 25 μL/min flow rate, a nozzle temperature of 37.5 °C, four cycles at 3 mm track spacing with a crisscross pattern, a 2 s drying period, 1,200 mm/min spray head velocity, 10 PSI of nitrogen gas, and a 40 mm nozzle height. After OTCD, DHB Sigma- at a concentration of 40 mg/mL in 70% MeOH and was sprayed at a 50 μL/min flow rate using the same M5 Sprayer. The nozzle temperature was set to 70 °C, with 12 cycles at 3 mm track spacing with a crisscross pattern. A 2 s drying period was added between cycles, and a linear flow was set to 1200 mm/min with 10 PSI of nitrogen gas and a 40 mm nozzle height. This resulted in a matrix coverage of ~667 μg/cm2 for DHB. Analyses were performed on a 12 T solariX FTICR MS, equipped with a ParaCell and an Apollo II ESI and MALDI source with a 2 kHz SmartBeam II frequency-tripled (355 nm) Nd: YAG laser (Bruker Daltonics, Bremen, Germany). Positive ion mode acquisitions were acquired with 100 laser shots per pixel at 1 kHz with 25 μm SmartWalk, where laser power was optimized prior to each acquisition and imaging was completed at a 25 μm step size. Broadband excitation from m/z 98.3 to 1000 was used, resulting in a detected transient of 0.5593 s with an observed mass resolution was ~110k at m/z 400 with lock mass to the molecular ion of 4-APEBA. Data collected with FlexImaging (v4.1; Bruker Daltonics, Bremen, DE) was imported into SCiLS Lab (v2024b, Bruker Daltonics, Bremen, DE) where centroided datasets were created for annotation by METASPACE. This open cloud software platform performs anno-tation based not only on the accurate mass information but also on a comprehensive bioinformatics framework that considers the relative intensities and spatial colocalization of isotopic peaks and also quan-tifies spatial information with a measure of spatial chaos followed by estimation of the FDR[72]. Datasets were annotated with sub-3 ppm mass error and searched with the possible chemical modification of [+C18H22N2Br]. Data were annotated using CoreMetabolome - v3 database and are reported with an FDR of ≤20%.

Lipidomic imaging was performed after deposition of DHA at a concentration of 15 mg/mL in 90% ACN with 0.2% TFA with a M5 Sprayer (HTX Technologies, Chapel Hill, NC), the matrix supernatant was sprayed after sonication and centrifugation. The following spraying parameters were used: a 150 µL/min flow rate, a nozzle temperature of 30.0 °C, four cycles at 2 mm track spacing with a crisscross pattern, 1300 mm/min spray head velocity, 10 PSI of nitrogen gas, and a 40 mm nozzle height. This resulted in matrix coverage of ~277 µg/cm2. Analyses were performed on a research grade Thermo Scientific Q Exactive Orbitrap MS, this platform was equipped with a MALDI source (Spectroglyph LLC, Kennewick, WA) with a 2 kHz Explorer One (349 nm) laser (Spectra Physics, Stahnsdorf, Germany)[73]. This platform was upgraded with ultrahigh mass range (UHMR) Q Exactive boards and operated under custom privileges as described in greater depth elsewhere[74]. Briefly, positive ion mode acquisitions were acquired with 1000 laser shots per pixel at 1 kHz with an average pulse energy of ~1.20 µJ, this resulted in a spot size measured at roughly 12 by 15 µm and imaging was completed at a 15 µm step size. The instrument was set for detection from m/z 600 to 2000, with a transient length of 1.024 s resultant in an estimated resolution of 480k at m/z 200 with an observed mass resolution of ~260k at m/z 804. In source trapping was disabled, with ion transmission and m/z detection set to low. Raw datasets were combined with.xml coordinates in SCiLS Lab (v2021c; Bruker Daltonics, Bremen, DE) and centroided datasets were exported from Mozaic (v2023.4.0.b3; Spectroswiss, Lausanne, CH) for annotation by METASPACE where datasets were annotated with sub-3 ppm mass error. Data were annotated using LipidMaps-2017-12-12 database and reported with an FDR of ≤20%.

## Histological imaging
Four post-MALDI placenta sections on ITO slide and three placenta sections on PEN slides were H&E-stained. After metabolomic imaging, matrix was removed by submerging placenta sections in two changes of 70% MeOH for 1 min. After lipidomic imaging, matrix was removed by submerging placenta sections in two changes of 70% ACN for 1 min. All sections were hydrated with 96% ethanol following 70% ethanol for 30 sec, and water for 1 min, and then H&E-stained per manufacturer protocol.

## Spatial proteomics analysis
**Laser capture microdissection (LCM).** Previously identified metabolically enriched villous subregions (STB and core) were microdissected using a PALM MicroBeam system (ZEISS) and collected in the corresponding microwells of the microPOTS chip that was preloaded with 2 µl of DMSO, which served as a capturing medium for excised tissue voxels. We collected fourteen replicates of STB subregions and thirteen replicates of corresponding core subregions, with each replicate containing 10,000 µm$^2$ tissue area. Metabolome-informed subregions mapped on sections 18 and 22 were collected from three adjacent placenta sections (sections 19, 23, and 26), profiling a tissue depth of 96 µm (Supplementary Fig. 3).

**Proteomics sample processing in a microdroplet and LC-MS/MS peptide analyses.** The microPOTS chip and its cover were incubated at 75 °C for 1 h to dry the DMSO solvent. Next, 2 µl of extraction buffer containing 0.1% DDM, 0.5×PBS, 50 mM TEAB and 1 mM DTT was dispensed into each well of the chip. The chip was incubated at 75 °C for 1 h. Thereafter, 0.5 µl of IAA solution (10 mM IAA in 100 mM TEAB) was added to the corresponding wells with the samples, followed by incubation at room temperature for 30 min. All samples were subsequently digested by adding 0.5 µl of an enzyme mixture (10 ng of Lys-C and 40 ng of trypsin in 100 mM TEAB) and incubated at 37 °C for 10 h. Following digestion, peptides were acidified by adding 5% FA to each sample to a final concentration of 1% FA. Each sample was collected and dispensed into a 20-µl aliquot of LC buffer A (water with 0.1% FA),

centrifuged at 10,000 g for 5 min at 25 °C and transferred (~22 µl) to an autosampler vial coated with 0.01% DDM. To minimize droplet evaporation, during every manipulation of the sample, the microPOTS chip was placed on an ice pack. Also, during each incubation, the microPOTS chip was sealed with the chip cover, wrapped in aluminum foil and incubated in a humidified chamber.

Proteomic analyses were performed using a Vanquish Neo UHPLC system (Thermo Scientific) coupled to an Orbitrap Exploris 240 Mass Spectrometer (Thermo Scientific). The sample was fully injected into a 20-µl loop and loaded onto an SPE precolumn (150 µm inner diameter (i.d.), 3 cm length) using Buffer A at a flow rate of 5 µl min−1 for 6 min. The SPE precolumn was slurry packed with 5-µm Jupiter C18 packing material (300-Å pore size; Phenomenex). Following SPE cleanup, the concentrated sample was backflushed on an LC column (75 µm i.d., 30-cm, 1.7 µm particle size, Waters BeH130). Chromatographic separation was performed at 200 nl min−1 using the following gradient: 1–8% (0 min–7 min), 8–21% (7 min–82 min), 21–28% (82 min–102 min), 28–37% (102 min–112 min), 37–75% (112 min–117 min) and 75%-95% (117 min–120 min) Buffer B (0.1% FA in acetonitrile) followed by column washing and re-equilibration. Separated peptides were introduced to the ionization source in which high voltage (2200 V) was applied to generate electrospray and ionize peptides. The ion transfer tube was heated to 300 °C, and the S-Lens RF level was set to 60. Data were acquired in positive mode. Full MS scans were acquired across a scan range of 300 to 1800 m/z at a resolution of 60,000, with automatic gain control (AGC) set to standard and maximum ion injection time set to auto. The intensity threshold was set to 5.0e3. The most intense ions selected in the first MS scan were isolated for higher-energy collision-induced dissociation (HCD) at a precursor isolation window width of 0.7 m/z, and normalized AGC of 250%. Maximum ion injection time for MS2 was set to auto with MS2 resolution set to 30,000. To improve the confidence of our peptide-to-spectrum matches, we leveraged Orbitrap-based high resolving power and mass accuracy in both MS and MS/MS analyses. The first mass and the normalized collision energy were set to 110 m/z and 30%, respectively. Proteomics data were searched against human proteome (UniProt release 2023_01) using MS-GF+ software (PMID: 25358478). The MS1-level area under the curve AUC intensities were obtained by MASIC software (PMID: 18440872). The peptide identifications were filtered by 5 ppm mass accuracy and 1% false discovery rate. Since acquired data contained high-resolution MS and high-resolution MS/MS spectra, proteins with at least 1 peptide-spectra-match (PSM) were included for downstream data analysis. All proteins discussed in this paper are listed in Supplementary Data 10. All proteins with only 1 peptide that were identified in at least 2 samples were manually inspected, and their annotated mass spectrum is reported in Supplementary Fig. 2.

## Statistical methods
Peak intensities were log2 transformed, and all missing intensity values were converted to NA values. Peptides without enough observations to conduct a statistical test, quantitative or qualitative, were filtered from the data[75]. A robust Mahalanobis distance based on peptide abundance vectors (rMd-PAV) was calculated to identify potential sample outliers in the data[76] using a p-value of 0.0001; one flagged sample was removed after being determined to be an outlier confirmed by principal component analysis (PCA). The Statistical Procedure for the Analyses of peptide abundance Normalization Strategies (SPANS) was implemented and determined the optimal normalization method to be global median centering without introducing bias into the data[77]. A test for a difference in mean abundance between STB and core regions was conducted while accounting for the non-independence of peptide and tissue section. For each protein, data for all peptides mapping to the protein were analyzed simultaneously. A linear mixed effects model was fit to the data, for each protein, with a fixed effect for region type (STB or core) and random effects for peptide and tissue section.

Additionally, a generalized linear mixed effects model was fit to the data with a conditional binomial distribution. The detection of a peptide was coded as a 0/1 as the dependent variable, and the model included a fixed effect for region type and random effects for peptide and tissue section. A chi-squared likelihood ratio test was conducted to test for a difference in the probability of observing a protein while accounting for the non-independence of peptide and tissue section. A Benjamini-Hochberg[78] was used to correct for multiple comparisons for both models. Detailed statistical report in Supplementary Statistical Methods.

## Enrichment analysis

Functional enrichment analyses were performed for the proteins that were differentially expressed or their abundances were increased in STB and core. The proteins with ANOVA adjusted $p$-values < 0.05 or G test adjusted $p$-values < 0.05 were subjected to functional enrichment. GO annotations[79], KEGG annotations[80], and HumanCyc annotations[81] were used to identify biological processes and metabolic pathways associated with STB and core. The GO term enrichment or KEGG module enrichment was performed in R using the clusterProfiler package[82], and the HumanCyc pathway enrichment was performed on humancyc.org using SmartTables[83].

## Multi-omics data integration

After identifying metabolomic pathways that were overrepresented in STB and core, we integrated our multi-omics data to confidently reconstruct pathways related to fatty acids and steroid processing. We manually paired colocalized enzymes with metabolomic evidence to illuminate the conversion reaction of the certain pathways, following the pathway reconstruction using enzymes mapped as differentially expressed or increased in STB and core. As depicted in our Supplementary Fig. 5, we used our multiomics data to reconstitute the following reference pathways: KEGG Fatty acid degradation (map00071), KEGG Steroid hormone biosynthesis (map00140), KEGG Sphingolipid metabolism (map00600), and HumanCyc - Pathway ketolysis. All pathways discussed in our manuscript are accompanied by citations referencing previous studies that reported on tissue-specific pathways.

## Reporting summary

Further information on research design is available in the Nature Portfolio Reporting Summary linked to this article.

## Data availability

MALDI MSI data generated for lipidomic and metabolomic imaging can be found at: Section 1 – lipidomic imaging https://metaspace2020.eu/annotations?db_id=24&ds=2024-07-12_21h29m37s&fdr=0.2&row=15 Section 2 – lipidomic imaging https://metaspace2020.eu/annotations?db_id=24&prj=0b2d571a-42e1-11ef-86c2-4b75175175b6&ds=2024-07-12_21h26m04s&fdr=0.2&page=2&row=4 Section 3 – metabolomic imaging https://metaspace2020.eu/annotations?db_id=38&ds=2024-02-26_22h34m46s&fdr=0.2&sort=fdr_msm Section 4 – metabolomic imaging https://metaspace2020.eu/annotations?db_id=38&prj=0b2d571a-42e1-11ef-86c2-4b75175175b6&ds=2024-02-26_22h36m04s&fdr=0.2&row=9 The Raw proteomics, lipidomics, and metabolomics data have been deposed to Mass Spectrometry Interactive Virtual Environment (MassIVE) (https://massive.ucsd.edu) and can be accessed with dataset identifier MSV000095456 for MassIVE and PXD054289 for ProteomeXchange. Source data are provided with this paper.

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

## Acknowledgements

The authors thank graphic designer Nathan Johnson (PNNL) for preparing the figures. This research is affiliated with the Pacific Northwest BioMedical Innovation Co-laboratory (PMedIC) joint research collaboration between OHSU and PNNL, conducted under the Laboratory Directed Research and Development Program at Pacific Northwest National Laboratory, a multi-program national laboratory operated by Battelle for the Department of Energy. This research utilized capabilities developed by the US Department of Energy, Office of Science, Biological and Environmental Research program Early Career Research Program Award to K.E.B.-J. Pacific Northwest National Laboratory is a multi-program national laboratory operated by Battelle for the DOE under Contract No. DE-AC05-76RL01830.

## Author contributions

K.E.B.-J. conceived and supervised this work. K.E.B.-J. proposed and M.V. designed the experiment. L.M. and L.K. provided tissue samples for this research and helped with histological guidance. K.J.Z. performed lipidomic imaging. D.V. performed metabolomic imaging. M.V. analyzed metabolomics and lipidomics data. M.V. performed cryosectioning, laser microdissection, and proteomics sample preparation. S.M.W. provided support for spatial proteomics capability. T.L.F., D.O., and R.J.M. analyzed proteomics samples. M.E.M. set up MS-GF+ search and uploaded the data to the repository. Y.G. performed proteomics data analyses. L.M.B. performed statistical analyses. J.K. performed an enrichment analysis. M.V. integrated multi-omics data and carried out biological data interpretation with L.K. contribution. R.W. and P.D.P. provided insightful comments about data interpretation. M.V. wrote the manuscript, K.E.B.-J. finalized it, and all authors provided their input and edits.

## Competing interests

The authors declare no competing interests.

## Additional information

**Marija Veličković** ® **¹, Leena Kadam², Joonhoon Kim** ® **³, Kevin J. Zemaitis** ® **¹, Dušan Veličković¹, Yuqian Gao** ® **⁴, Ruonan Wu** ® **⁴, Thomas L. Fillmore¹, Daniel Orton** ® **⁴, Sarah M. Williams¹, Matthew E. Monroe** ® **⁴, Ronald J. Moore** ® **⁴, Paul D. Piehowski** ® **¹, Lisa M. Bramer** ® **⁴ ✉, Leslie Myatt² ✉ & Kristin E. Burnum-Johnson** ® **¹ ✉**

¹The Environmental Molecular Sciences Laboratory, Pacific Northwest National Laboratory, Richland, WA, USA. ²Department of Obstetrics & Gynecology, Oregon Health & Science University, Portland, OR, USA. ³Energy and Environment Directorate, Pacific Northwest National Laboratory, Richland, WA, USA. ⁴Biological Sciences Division, Pacific Northwest National Laboratory, Richland, WA, USA. ✉e-mail: lisa.bramer@pnnl.gov; myattl@ohsu.edu; Kristin.Burnum-Johnson@pnnl.gov

