## [Transparent Peer Review file · Nature Communications]

Advanced multi-modal mass spectrometry imaging reveals functional differences of placental villous compartments at microscale resolution

Corresponding Author: Dr Kristin Burnum-Johnson

Version 0:

Reviewer comments:

Reviewer #1

(Remarks to the Author)

Interesting article mainly building upon their earlier research.

* Although that earlier work is mentioned a few times in the paper, I miss the specific reference making the original work hard to find. Also, it would be good to make it more clear what exactly is much different besides the biological interpretation.

* The method used is not very new and misses some crucial reference articles doing multi-omics in a very similar way. Even on the same single section in some cases. So method wise I am not convinced this is Nature communication material.

* Check the reference list since some are missing the authors

* using a 25um pixel size MSI is in my opinion not detailed enough to draw the biological conclusions as you do in the paper. What is the size of the SBT cell? In addition to that, add scale bars in all images. That put things in perspective

* Add H&E stainings to show the pathological validations of the areas.

* Looking at the Metaspace data on metabolomics, there is something wrong going on since there are masses detected outside the tissue. Can you comment on that.

* I miss in the material and methods the detailed information how you connect the proteomics to the lipidomics and metabolomics data. Which platforms did you use for pathway analysis?

* On the LCM method: what is the area of the ROI's you sectioned? Can you comment on that? Based on what exactly did you define these ROI's to cut out? lipidomics or metabolomics? And how did you do this in practice to have detailed LCM of the correct regions?

* in general the introduction is too long compared to the research work that could use some more explanations to be able to reproduce the paper.

Reviewer #2

(Remarks to the Author)

Velickovic et al. present a spatial multi-omics study of placental villous where adjacent sections were analyzed with either MALDI-imaging mass spectrometry enhanced with On-Tissue Chemical Derivatization (OTCD) for spatial metabolomics, or with spatially-resolved proteomics by laser-capture microdissection (LCM) followed by highly-sensitive proteomics.

The dataset collected in this study is of high interest in the field of spatial multi-omics, because it provides a high metabolite coverage (because of the use of OTCD) and untargeted proteomics for the LCM'ed areas. This dataset can represent a useful resource both for computational method developers, as well as for biologists interested in placenta.

However, there are several major comments which need to be addressed if the manuscript will be selected for revision.

*** Major Comments ***

i. In Fig 2, it is not clear that region defined by metabolomic imaging is used for spatial proteomics in the adjacent section (not highlighted in the figure). In addition, it's also not clear how comparative analysis between STB and core is used alongside metabolites for pathway integration. It would also be nice to provide a schematic explaining the main concepts behind the pathway integration strategy either in main figure or supplementary.

ii. I acknowledge the significant effort invested in the experimental design and data acquisition. However, the representation of the statistical analysis is lacking. There is limited information regarding the assumptions behind the methods used, such as ANOVA, G-test, and mixed effect models. The authors should offer more comprehensive details on the statistical approaches for both proteomics and metabolomics individually, as well as provide a more thorough explanation of the integrative pathway analysis, which is also insufficiently addressed.

iii. In the results section entitled “Proteome profiling of metabolome-informed placental villi subregions”, multiple proteins were provided as relevant in either STB or Core cells and seems to be associated with various biological processes. However, there are no figure or tables to summarize such results and whether these processes are significantly overrepresented. Maybe an enrichment analysis of such differentially expressed proteins would shed some light on such processes. Or at least a simple visualization to associate those proteins with relevant processes and how are they different between STB and Core.

iv. I agree with the authors that using co-detection provided spatial context and can be performed in an unbiased way. I also agree that co-detection is a straightforward initial step in spatial multi-omics integration and to understand the likely hotspot of pathway activity which helps generate hypotheses about such pathways and their activity. However, there are several limitations with just using co-detection to get pathway-level insights :

1. Co-detection doesn't account for how molecules interact. So simply detecting an enzyme and its substrate in proximity doesn't imply any functional interactions or co-regulation.
2. Co-detection as presented in this study doesn't take into account the relative abundance of the proteins and metabolites in the region which is critical for understanding pathway activity.
3. Co-detection cannot distinguish between pathway cross-talk. Co-detected molecules could simply result from an overlap of multiple pathways.
4. Spatial proximity doesn't always imply functional relevance.

Please mention some of these limitations and be more critical of what pathway-level insights can be obtained from co-detection. For example, in lines 293-294 the authors state that “Fatty acid binding proteins 4 and 5 (FABP4 and FABP5), detected in both villous compartments, further indicate that intracellular FFAs are bound and subsequently processed to membrane phospholipids”. Such claims about interactions and processing cannot be inferred from co-detection, it can only be hypothesized based on proximity. So please make sure not to provide oversimplified interpretations and acknowledge such limitations in those claims.

v. The pathways that the authors reference in Fig 5 are not cited . For example, in Fig 5b, the authors highlight few reactions from the “estrogen synthesis pathway”, which resource was used (e.g. KEGG, Reactome) or if it's curated from literature, where are the corresponding citations. If it is retrieved from a pathway databases, these reactions has to be viewed as part of the larger pathway entry to provide a better contextual view of the regulatory interactions involved. Checkout out PathView (<https://pathview.uncc.edu/>) for examples.

vi. In the discussion, the authors state that “integration multi-omics data can enhance confidence in lipid and metabolite identifications by colocalizing the detection of corresponding synthetic enzymes”. This is an oversimplification of metabolite/lipid identification because, as discussed before, co-detection can suggest functional relationship, but doesn't guarantee one. The presence of a metabolite and its synthetic enzyme doesn't confirm that the enzyme is actively synthesizing that metabolite at that location, because of many reasons. One being that some enzymes could generate structurally similar molecules as they can be non-specific. Other reasons include : post-translation regulation (expression vs activity) and transport of metabolites/lipid (synthesis site is different from activity site).

vii. While the suggested modifications seem to improve the MIPi workflow experimentally, the data analysis performed seems rather primitive in comparison. A detailed characterization of metabolic and proteomics profile in selected sROIs are missing, also the spatial localization of the 14 selected ROIs on the adjacent sections are not reported. Other missing questions include :

1. How different are the STB/core regions on the same sections (intrareplicate variability) and in adjacent sections ?
2. How different are corresponding STB-Core pairs in each sections?. For example Placenta_19_R1_core and Placenta_19_R1_STB vs Placenta_19_R2_core and Placenta_19_R2_STB.
3. How are the features correlated across the regions and sections. Metabolite-metabolite or protein-protein correlations ?
4. Are the profiles observed in STB-Core regions preserved across adjacent sections ?

Additional integrative analysis could also be performed given the provided data that could use relative intensity of the molecules in addition to just using co-detection.

viii. The paper and workflow in Figure 2 claims multi-omics data integration performed. This reads like an overstatement, as there is no integration performed. The authors have performed analysis of each omics dataset separately, and then manually picked metabolites and enzymes from the same pathways. However, normally data integration refers to an automated procedure.

*** Minor Comments ***

i. In line 68, the sentence “investigating morphological and molecular changes in the placenta could transform our discoveries in personalized therapies and novel therapeutics in maternal-fetal medicine” is a generic statement. Try to be more specific or use fewer generic statements.

ii. In lines 69-72, the authors state in the introduction that only bulk samples were used in different omics studies. However, there are other spatial and/or single cell omics studies on the placenta that should be acknowledged as well. Please consider the following references :

1. <https://www.nature.com/articles/s41586-023-05869-0>
2. <https://www.nature.com/articles/s41588-023-01647-w>

iii. In Lines 77-79, the authors state that “examining cellular diversity and tissue heterogeneity without dissociating the cells and disrupting their direct interactions can be done by mass spectrometry imaging (MSI)”. This applies to most if not all

spatial omics technologies and is not exclusive to MSI. Please adjust accordingly.

iv. In line 110, use either omics or modalities

v. Add citation of previously published MIP1 workflow in lines 120-123.

vi. Providing data as a pdf table makes it harder to reproduce the analysis. Please provide it as a spreadsheet so that it would be easier to reanalyze.

vii. In Fig 3a, it's hard to see the distinction between STB and Core, would it help if another image at better resolution and higher brightness is provided?

viii. In lines 169-171, the authors mention differential expression patterns of specific lipid classes. Where is the corresponding figure/table of such classes?. And which of the measured lipids are annotated to those classes?.

ix. In line 172, the authors mention that "stem villus region was enriched with lipid molecules that colocalized mainly with the villous core". How was the enrichment calculated?. If no enrichment analysis was performed, please replace enriched with X lipid molecules were detected in high abundance (intensity > X) in X region compared with Y region. Also, how was colocalization calculated?. If the authors use the colocalized molecules reported by METASPACE, please specify.

x. In line 205-206, the authors mentioned that proteome profiling of 14 villous regions were performed informed by previous metabolomic imaging. I don't see any reference to those 14 regions in a figure or a table and their spatial localization on the respective tissues. Can you please add a reference to those regions?. And if they were metabolome informed, why are they not discussed in the previous results sections concerning metabolomics results?

xi. In line 209, those proteins are differential expressed, not enriched based on the ANOVA test. Please rephrase accordingly.

xii. Some figures are cited as (Fig X) as in line 207 and others as (Figs. X) as in line 211. Please use consistent in-text citations.

Reviewer #3

(Remarks to the Author)

Reviewer #4

(Remarks to the Author)

This paper by Veličković et al demonstrates an impressive spatial metabolomic and proteomic workflow demonstrated on placental tissue. For a spatial approach they obtain high level of sensitivity for lipids and metabolites measured. The examples where the approach was able to identify substrate enzyme and product in the same region provides confidence in the results and demonstrate their utility. The workflow involves technical advances to increase sensitivity e.g. on-tissue chemical derivatization.

The paper uses the term "tentatively analyzed as" due to the technical limitations in annotating compounds. While caution is the correct approach here, this is nevertheless a limitation. How might a higher degree of certainty in the molecular annotation be achieved through technical innovation or combining data from other sources.

The distinction between trophoblast and core might not be as ambitious as it could be especially as regards the core which contains the greatest cellular heterogeneity. In figure 3 there are structures that appear to be vessels from the histology and these could be easily identified as distinct regions?

The histology images, at least in the downloaded file, do not appear to be of high quality and I wonder if these could be improved? Details of the microscopy are not reported but the use of a slide scanner would produce high resolution tiled image of the whole structure. Also, could the current images be improved by adjusting the brightness?

In figure 3 the expanded region could be shown with and without the overly so the underlying histology could be better understood, or at least a box placed on the histology image to show where it comes from.

This study represents a significant increase in resolution compared to previous work, but the holy grail would be a true cellular analysis (for instance separating syncytiotrophoblast and cytotrophoblast). The discussion could have included a more comprehensive limitations section addressing current limitations (including confidence in molecular annotation) and outlining those limiting future progress such as further increases in resolution.

Could the authors comment further on the potential for this approach to be used to compare healthy and pathological tissue?

As a minor points:

Line 287, rather than failure to colocalize could in this region a higher flux from substrate to product could result resulting in levels of substrate lower than the level of detection.

The syncytiotrophoblast is a syncytium and I would not refer to STB cells.

The figure 1 legend could be reworded slightly to make clear that the syncytiotrophoblast is not a layer of cells but a syncytium.

Version 1:

Reviewer comments:

Reviewer #1

(Remarks to the Author)

Although the authors took a lot of effort to answer the many comments of all reviewers, I still do not see this paper as Nature

Communication material since:

- * The main workflow part is not new at all and having a higher amount of proteins from extraction is not enough for a high impact paper. The number of tissues should also increase to state their conclusions confidently.
- * Although a very high mass resolution MS is used, there are still tentative ID's which raises questions. MS/MS analysis could have been done
- * the spatial resolution used is not high enough to draw the conclusions they do from their data. This also includes the precision of the LCM and the H&E now added (not convincing at all to do LCM on such a small area).

Reviewer #2

(Remarks to the Author)

The authors have performed an extensive revision of the text and figures which substantially improved the manuscript.

I believe that my concerns were substantially addressed.

Reviewer #3

(Remarks to the Author)

Reviewer #4

(Remarks to the Author)

The manuscript has been clarified and improved in response to the referees comments. The points I raised have been largely addressed. I do not have any further substantial comments.

some inset higher refs images might be nice in figure 5c

NCOMMS-24-50998-T

We thank all reviewers for the insightful comments, which have improved the clarity and detail of our manuscript. In this study, we aim to demonstrate key advancements in previously published analytical workflows for deep multi-omic imaging of distinct microscale tissue regions. Our approach generates metabolite-protein interaction-specific images and integrates them to provide pathway-level resolution tailored to unique placental tissue microenvironments.

We made a few minor amendments to our manuscript that did not change any results, interpretations, or conclusions. We thank the reviewers for their comments, which led to the discovery of these necessary changes.

- We have revised our Figure 4c to include a new volcano plot for the G test. An error in the initial version of our manuscript affected only the volcano plot graphic, not the data. The results discussed in our initial manuscript are now consistent with the corrected Figure 4c. We apologize for including the incorrect volcano plot in our initial submission.

Revised Fig. 4 as it appears in the manuscript:

Fig 4. Unveiling subregion-specific enzymes using microPOTS processing. a) LCM collection of placenta villous subregions. b) Volcano plot for the abundance-based model comparing the tissue region means for each protein. c) Volcano plot for the probability of detection-based model comparing the tissue region mean detection probabilities for each protein. Red hexagons and blue hexagons represent proteins identified exclusively in the STB and Core, respectively. The color of each hexagon corresponds to the scale bars below the graph, which indicate the number of proteins (N). In the present volcano plot, extreme values were not shown. The plot with results for all proteins can be found in Figure 12 of Supplementary Statistical Methods.

Figure 12 of Supplementary Statistical Methods. Volcano plot for the probability of detection-based model comparing tissue region mean detection probabilities for each protein. Red hexagons and blue hexagons represent proteins identified exclusively in the STB and Core, respectively. The color of each hexagon corresponds to the scale bars below the graph, which indicate the number of proteins (N). The plot with results for all proteins.

- In our revised manuscript, we adjusted the FDR reporting for metabolite imaging. Although all metabolites used for pathway reconstruction were annotated with an FDR of $\leq 10\%$, we increased the FDR here for the sake of consistency with lipidomic imaging and previous efforts from our group to $\leq 20\%$. We believe it is important to include these additional ion images in our supplemental data file as they provide a valuable resource for researchers in future studies.

Revision as it appears in the manuscript:

By leveraging the METASPACE annotation platform to search against the CoreMetabolome database, we tentatively annotated ~~308~~ 716 and ~~356~~ 618 ion images from two adjacent sections, respectively, at ~~10-20%~~ FDR (Supplementary Tables ~~3 & 4~~ 4 & 5).

- In the revised manuscript, we have also included enrichment analysis and provided comprehensive details on the functional and pathway integration of our multi-omics data which leveraged the Gene Ontology (GO), the Kyoto Encyclopedia of Genes and Genomes (KEGG) and the Encyclopedia of Human Genes and Metabolism (HumanCyc) databases.

See detailed remarks to authors below.

Reviewer #1 (Remarks to the Author)

Interesting article mainly building upon their earlier research.

We are grateful to the reviewer for taking the time to review our article, and for their insightful comments that improved the clarity and detail of our manuscript. We have addressed each comment as detailed below.

* Although that earlier work is mentioned a few times in the paper, I miss the specific reference making the original work hard to find. Also, it would be good to make it more clear what exactly is much different besides the biological interpretation.

We thank the reviewer for this important comment. We revised our manuscript to reference our original MIPI work in the introduction.

The MIPI approach described in our manuscript goes beyond histologically guided identification of regions of interest (ROIs) and utilizes spatial metabolite data to inform the subsequent selection and laser capture microdissection (LCM) of discrete placental functional units. The advancements of our MIPI workflow are discussed throughout our manuscript, and below are some of the **key differences**:

- We **advanced our metabolomic imaging by implementing on-tissue chemical derivatization (OTCD)**, which enabled us to boost sensitivity and detect endogenous steroid hormones that are challenging to detect with conventional Matrix-Assisted Laser Desorption/Ionization (MALDI) – Mass Spectrometry Imaging (MSI) workflows due to their low ionization efficiency. For OTCD, we used a combined application of EDC and 4-APEBA, which has previously been demonstrated to surpass the benefits of conventional derivatization agents used for carboxylic acid and aldehyde/ketone derivatization. Our advanced workflow ensured that the free fatty acids (FFAs) detected and discussed throughout our manuscript are endogenous. All FFAs were captured in their derivatized form; hence, discussed FFAs were not fragments of lipids artificially generated by the MALDI laser.
- Complementary **lipidomic imaging was incorporated** to expand detection coverage, providing more comprehensive data.
- We **included histological hematoxylin and eosin (H&E) staining in our MIPI workflow** to better visualize morphology and correlate with molecular changes. All post-MALDI sections were H&E-stained and overlaid with ion images to visualize morphology, correlate morphology with molecular changes, and map the areas for subsequent proteome profiling. Additionally, all sections for microdissection were H&E-stained to ensure precise mapping of areas, identified by our spatial metabolite data.
- We **enhanced the spatial resolution of our MALDI-MSI measurements by using 25 μm and 15 μm step sizes for metabolomic and lipidomic imaging**, respectively, compared to the previously used 50 μm spatial resolution. Improved spatial resolution allowed us to resolve molecules across the placental tissue section with finer details and characterized the subtle subregional differences of villous functional units (STB and core), guiding subsequent subregional collection.
- Leveraging our microdroplet processing platform, we reliably analyzed subregional differences in protein levels within villous compartments. Consequently, **we demonstrated improved sensitivity in our proteomics imaging** by collecting and processing smaller tissue amounts (10,000 μm^2 tissue area) compared to our previous MIPI experiments, which profiled regions ranging from 400,000 to 700,000 μm^2 in area.

- **Pathway enrichment analysis** provided insights into significantly overrepresented pathways in each of the villous subregions, allowing us to identify and map specific metabolic pathways and reconstruct them using our multi-omics data.

* The method used is not very new and misses some crucial reference articles doing multi-omics in a very similar way. Even on the same single section in some cases. So method wise I am not convinced this is Nature communication material.

We thank the reviewer for their comment. While our manuscript focuses on references specifically relevant to omics analyses of placenta tissue, we have revised our manuscript to include references to previous spatial multi-omics publications. To address the reviewer's comment, we discussed several references from previously published spatial omics studies.

We agree that these added references include similar workflows with respect to the objective – the acquisition of multi-omics data. However, in the following paragraph, we detail the unique benefits of our MIPI approach compared to these other methodologies to justify why we submitted our manuscript to Nature Communication.

Previous workflows rely on histology-guided LCM collection, they do not use metabolome-guided proteome profiling. Although Hendriks et al. [1] presented an interesting and potentially very useful approach of an MSI-guided lipidomics and proteomics workflow from a single section of glioblastoma multiforme brain tumor, it lacks the relevant information such as MALDI-MSI images and metabolite data that can support their claims. Claes et al. [2] previously published a MALDI-IHC-guided spatial proteomics workflow on breast cancer tissue which uses targeted MALDI-IHC imaging to guide bottom-up spatial proteomics. Although it is promising with regard to the further development of the spatial omics field, it focuses only on proteomics, hence it was not referenced in our paper. Also, Pace et al. [3] utilized a very similar multi-omics workflow on rat brain tissue, which combines infrared matrix-assisted laser desorption electrospray ionization (IR-MALDESI) for metabolomic MSI and nanodroplet processing in one pot for trace samples (nanoPOTS) LC-MS/MS for spatially resolved proteome profiling. They acquired some MALDI-MSI datasets, but in their approach, they utilized histology-guided LCM collection. In regard to metabolic imaging, they used lateral resolution of 150 μm , compared with our lipidomic and metabolomic imaging that were done with 15 μm and 25 μm step size, respectively. The size of the collected tissue pixel for spatial proteomics and obtained depth of proteome coverage were similar. However, it is important to note that proteome coverage should not be directly compared between different tissue types and analytical instruments. Additionally, in their study, they reported only one instance of a multi-omics integration effort, which involved using a protein and its corresponding substrate/product across different ROIs. The presented multi-omics technology utilized similar MS modalities; however, we would like to reiterate that MIPI is uniquely focused on profiling the hotspots of specific pathway activities. This targeted approach allows for comprehensive biological interpretation and has the potential to address specific biological questions. Mezger et al. [4] presented a very interesting multi-omics approach that utilize MSI-guided proteomics from a conductive, nonmembrane slide. Although this publication showed a promising approach, reported proteome coverage was significantly lower compared to coverage obtained from PEN membrane slides for both frozen and FFPE tissues.

[1] <https://pubs.acs.org/doi/10.1021/acs.analchem.3c05850> (now reference 44 in our manuscript)

[2] <https://pubs.acs.org/doi/10.1021/acs.analchem.2c04220>

[3] <https://pubs.acs.org/doi/10.1021/acs.jproteome.1c00641> (now reference 46 in our manuscript)

[4] <https://pubs.acs.org/doi/full/10.1021/acs.analchem.0c04572> (now reference 45 in our manuscript)

All the mentioned publications presented promising approaches to developing more comprehensive multi-omics technologies. However, compared to these approaches, they did not surpass the molecular coverage and lateral resolution that MIPI offers, nor did they provide the comprehensive biological interpretation necessary for addressing specific biological questions. To the best of our knowledge, the MIPI study was the first multi-omics MS-based imaging study on human placenta tissue that offered pathway-level resolution. Therefore, we believe that our work's advancements in technical approach, biological integration, and data interpretation make our manuscript well-suited for submission to Nature Communications.

* Check the reference list since some are missing the authors

We thank the reviewer for being diligent in their review. We have revised our reference list to include all authors.

* using a 25um pixel size MSI is in my opinion not detailed enough to draw the biological conclusions as you do in the paper. What is the size of the SBT cell? In addition to that, add scale bars in all images. That put things in perspective.

Our MIPI approach combined different omics modalities to comprehensively interrogate molecular dynamics within the full spatial context of the placenta tissue. In our study, we interrogated placental functional units, focusing on metabolic regions, not cells. We demonstrated that metabolomic imaging with 25 μm spatial resolution is sufficient to spatially resolve the molecule and characterize the subtle subregional differences between distinct villous compartments (STB and core) of placental villous functional units. Our spatial metabolite data was used to inform the subsequent selection and laser capture microdissection (LCM) of discrete placental villous subregions for complementary proteome profiling. Integration of our multi-omics data allowed us to reconstruct underlying biological pathways relevant to placental function across histologically distinct villi subregions.

As noted by another reviewer: "STB is not a layer of cells but a syncytium". Therefore, we revised our manuscript to make it clear that the syncytiotrophoblast (STB) is a syncytium.

We revised our manuscript to include scale bars in all our images.

* Add H&E stainings to show the pathological validations of the areas.

H&E-stained images were provided in the submitted manuscript.

* Looking at the Metaspace data on metabolomics, there is something wrong going on since there are masses detected outside the tissue. Can you comment on that.

Out of tissue annotations are present in our metabolomic imaging data because we analyzed surrounding tissue area as well, which is common practice in MSI analysis so that false positive annotations can be excluded. Those features are not endogenous molecules, they are background signal of MALDI matrix clusters, derivatization agent, impurities of the MALDI matrix, and ITO slide itself.

* I miss in the material and methods the detailed information how you connect the proteomics to the lipidomics and metabolomics data. Which platforms did you use for pathway analysis?

We agree that our multi-omics integration was not adequately explained. We revised our manuscript to include enrichment analysis and describe multi-omics data integration. We also included **Supplemental Figure 4** with reference pathways from KEGG and HymanCyc databases used for multiomics data reconstruction. All pathways discussed in our manuscript are accompanied by citations referencing previous studies that reported on tissue-specific pathways.

Revisions are throughout the manuscript text and the below paragraph is now in the methods.

Enrichment analysis. *Functional enrichment analyses were performed for the proteins that were differentially expressed or their abundances were increased in STB and core. The proteins with ANOVA adjusted p-values < 0.05 or G test adjusted p-values < 0.05 were subjected to functional enrichment. GO annotations[79], KEGG annotations[80], and HumanCyc annotations[81] were used to identify biological processes and metabolic pathways associated with STB and core. The GO term enrichment or KEGG module enrichment was performed in R using the clusterProfiler package[82], and the HumanCyc pathway enrichment was performed on humancyc.org using SmartTables[83].*

* On the LCM method: what is the area of the ROI's you sectioned? Can you comment on that? Based on what exactly did you define these ROI's to cut out? lipidomics or metabolomics? And how did you do this in practice to have detailed LCM of the correct regions?

Collected areas of the ROI's were 10,000 μm^2 , as stated in the submitted manuscript, under the section 'Laser capture microdissection (LCM)'.

Both, lipidomic and metabolomic imaging, provided molecular characterizations that outlined subtle subregional differences between the outer surface of the villous STB and the villous core. Unlike lipidomic imaging that showed comparable abundance across all imaged villus functional units, metabolomic imaging allowed us to map villous regions with enhanced metabolomic activities related to steroid and fatty acid metabolism, highlighting subtle subregional differences between the two compartments. Therefore, metabolomic imaging data defined discrete placental villous subregions for complementary proteome profiling. We revised our manuscript to include more details including the below explanation.

Proteome profiling of metabolome-informed placental villi subregions. *In contrast to lipidomic imaging, which showed comparable molecular abundance across all imaged villous functional units, advanced metabolomic imaging provided discerning molecular visualization, allowing us to pinpoint villous functional units with high metabolic activity related to fatty acid and steroid metabolism. Informed by the metabolome-specific villous regions (**Supplementary Fig. 3**), we complemented our metabolome data by profiling the proteomes of 14 villous regions (14 STB and 13 core subregions) from three adjacent sections (**Supplementary Tables 6, 7, and 8**).*

All post-MALDI sections on ITO slides were H&E-stained and overlaid with ion images using METASPACE, which allowed us to visualize morphology, correlate morphology with molecular changes, and map the sROIs for subsequent proteome profiling. Prior to microdissection, sections on PEN slides (allocated for spatial proteomics) were H&E-stained and overlaid with post-MALDI H&E-stained sections to precisely map the sROIs identified by our spatial metabolite data.

* in general the introduction is too long compared to the research work that could use some more explanations to be able to reproduce the paper.

Thank you for this important comment. We revised our manuscript to include more details in our experimental section.

Reviewer #2 (Remarks to the Author)

Velickovic et al. present a spatial multi-omics study of placental villous where adjacent sections were analyzed with either MALDI-imaging mass spectrometry enhanced with On-Tissue Chemical Derivatization (OTCD) for spatial metabolomics, or with spatially-resolved proteomics by laser-capture microdissection (LCM) followed by highly-sensitive proteomics.

The dataset collected in this study is of high interest in the field of spatial multi-omics, because it provides a high metabolite coverage (because of the use of OTCD) and untargeted proteomics for the LCM'ed areas. This dataset can represent a useful resource both for computational method developers, as well as for biologists interested in placenta.

However, there are several major comments which need to be addressed if the manuscript will be selected for revision.

We are grateful to the reviewer for taking time to review our article, and for their insightful comments that improved the clarity and detail of our manuscript. We addressed each comment as detailed below.

***** Major Comments *****

i. In Fig 2, it is not clear that region defined by metabolomic imaging is used for spatial proteomics in the adjacent section (not highlighted in the figure). In addition, it's also not clear how comparative analysis between STB and core is used alongside metabolites for pathway integration. It would also be nice to provide a schematic explaining the main concepts behind the pathway integration strategy either in main figure or supplementary.

We thank the reviewer for this important comment. We revised our Figure 2 to make it clear that we used adjacent sections for the subsequent selection and laser capture microdissection (LCM) of discrete placental villous subregions for complementary proteome profiling. We also included elements related to pathway integration in our Figure 2 and provided detailed explanation of the pathway integration in our manuscript.

Revised Fig. 2 as it appears in the manuscript:

Figure 2. Advanced MIPI approach. Schematic workflow of advanced MIPI approach that combines multi-modal MALDI-MSI for comprehensive lipidome and metabolome imaging with complementary microscale proteome profiling by microPOTS approach. 4-APEBA, 4-(2-((4-bromophenethyl)dimethylammonium)ethoxy) benzenaminium dibromide; DHA, 2,5-dihydroxyacetophenone; DHB, 2,5-dihydroxybenzoic acid.

ii. I acknowledge the significant effort invested in the experimental design and data acquisition. However, the representation of the statistical analysis is lacking. There is limited information regarding the assumptions behind the methods used, such as ANOVA, G-test, and mixed effect models. The authors should offer more comprehensive details on the statistical approaches for both proteomics and metabolomics individually, as well as provide a more thorough explanation of the integrative pathway analysis, which is also insufficiently addressed.

We thank the reviewer for this important comment and agree that more detail should have been included.

We revised our manuscript to provide additional information on the statistical analysis for proteomics data. We also included log₂ peptide data and normalized log₂ peptide data in our **Supplementary Tables 7 and 8**. We also provided a comprehensive report of our statistical analysis (**Supplementary Statistical Methods**) with details on the statistical approaches. Regarding the assumptions behind the used models, these details are provided in the “Model Assumptions” section of the **Supplementary Statistical Methods**.

*Model Assumptions section of the **Supplementary Statistical Methods** as it appears in the revised manuscript:*

For each protein, two models were fit to the data.

1. *Abundance model: When modeling the abundance response variable, a linear mixed effects model was used. Tissue region was included as a fixed effect. This model assumes: (a) error variances are*

normally distributed, (b) error variances are independent, and (c) variances are homoskedastic. Because multiple samples were taken from each tissue section and each protein (in almost all cases) has multiple peptides being analyzed, condition (b) does not hold. Thus, peptide and tissue section were included in the model to properly partition variation and ensure a valid statistical comparison of means from tissue region. Assumptions (a) and (c) are commonly assumed to be valid for log transformed mass spectrometry data.

- 2. Detection model: When modeling the probability of detecting a given protein, a generalized linear mixed effects model, with a conditional binomial distribution and a logit link function was used. Tissue region was included as a fixed effect. This model assumes: (a) error variances are independent. Because multiple samples were taken from each tissue section and each protein (in almost all cases) has multiple peptides being analyzed, condition (a) does not hold. Thus, peptide and tissue section were included in the model to properly partition variation and ensure a valid statistical comparison of means from tissue region.*

Regarding metabolomics data: The data were uploaded to METASPACE (<https://metaspace2020.eu>) for molecular annotation and data visualization. This open cloud software platform performs annotation based not only on the accurate mass information but also on a comprehensive bioinformatics framework that considers the relative intensities and spatial colocalization of isotopic peaks and quantifies spatial information with a measure of spatial chaos followed by estimation of the FDR*. Metabolomic data were annotated using CoreMetabolome database with an FDR of $\leq 20\%$. Although all metabolites used for pathway reconstruction were annotated with an FDR of $\leq 10\%$, we increased the FDR here for the sake of consistency with previous efforts from our group to 20%. We believe it is important to include these additional ion images in our supplemental data file as they provide a valuable resource for researchers in future studies. We revised our manuscript to capture these changes.

** Palmer, A. et al. FDR-controlled metabolite annotation for high-resolution imaging mass spectrometry. Nat. Methods 14, 57–60 (2017).*

We agree that multi-omics data integration was insufficiently addressed. Therefore, we have revised our manuscript to include a detailed explanation of the integrative pathway analysis throughout the results section and added the below methods paragraph and Supplementary Figure 5

Multi-omics data integration. *After identifying metabolomic pathways that were overrepresented in STB and core, we integrated our multi-omics data to confidently reconstruct pathways related to fatty acids and steroid processing. We manually paired colocalized enzymes with metabolomic evidence to illuminate the conversion reaction of the certain pathways, following the pathway reconstruction using enzymes mapped as differentially expressed or increased in STB and core. As depicted in our **Supplementary Figure 5**, we used our multiomics data to reconstitute the following reference pathways: KEGG Fatty acid degradation (map00071), KEGG Steroid hormone biosynthesis (map00140), KEGG Sphingolipid metabolism (map00600), and HumanCyc - Pathway ketolysis. All pathways discussed in our manuscript are accompanied by citations referencing previous studies that reported on tissue-specific pathways.*

iii. In the results section entitled “Proteome profiling of metabolome-informed placental villi subregions”, multiple proteins were provided as relevant in either STB or Core cells and seems to be associated with various biological processes. However, there are no figure or tables to summarize such results and whether

these processes are significantly overrepresented. Maybe an enrichment analysis of such differentially expressed proteins would shed some light on such processes. Or at least a simple visualization to associate those proteins with relevant processes and how are they different between STB and Core.

We thank the reviewer for this important comment. All the proteins discussed in our manuscript are listed in the **Supplementary Table 10** with their stats that indicate upregulation/presence in STB or core. We revised our manuscript to include the results of pathway enrichment analysis. We also included additional **Supplementary Fig. 4** with enrichment analysis results that clearly show functional differences between STB and core. Pathway enrichment analysis provided insights into significantly overrepresented pathways in each of the villous subregions, enabling us to map the certain metabolic pathways and reconstruct them using our multi-omics data. As such, steroid hormone biosynthesis, fatty acid beta-oxidation, and ketolysis were mapped as upregulated/increased, therefore they were reconstructed (**Supplementary Fig. 5**). Suggested *de novo* synthesis pathway of ceramide in the STB was not enriched due to the low number of significant proteins (2 out of 21), but it was curated from the literature and our lipidomic imaging data.

iv. I agree with the authors that using co-detection provided spatial context and can be performed in an unbiased way. I also agree that co-detection is a straightforward initial step in spatial multi-omics integration and to understand the likely hotspot of pathway activity which helps generate hypotheses about such pathways and their activity. However, there are several limitations with just using co-detection to get pathway-level insights :

1. Co-detection doesn't account for how molecules interact. So simply detecting an enzyme and its substrate in proximity doesn't imply any functional interactions or co-regulation.
2. Co-detection as presented in this study doesn't take into account the relative abundance of the proteins and metabolites in the region which is critical for understanding pathway activity.
3. Co-detection cannot distinguish between pathway cross-talk. Co-detected molecules could simply result from an overlap of multiple pathways.
4. Spatial proximity doesn't always imply functional relevance.

Please mention some of these limitations and be more critical of what pathway-level insights can be obtained from co-detection. For example, in lines 293-294 the authors state that "Fatty acid binding proteins 4 and 5 (FABP4 and FABP5), detected in both villous compartments, further indicate that intracellular FFAs are bound and subsequently processed to membrane phospholipids". Such claims about interactions and processing cannot be inferred from co-detection, it can only be hypothesized based on proximity. So please make sure not to provide oversimplified interpretations and acknowledge such limitations in those claims.

We agree with the reviewer's comment. We have made necessary modifications throughout the manuscript to reiterate to the reader about limitations of using co-detection to obtain pathway-level insights.

Revision as it appears in the manuscript: Fatty acid binding proteins 4 and 5 (FABP4 and FABP5), detected in both villous compartments, can suggest that intracellular FFAs are bound and subsequently processed to membrane phospholipids, used for energy production in mitochondria, or are esterified to triacylglycerols as the intermediate storage form in cells[62]. While spatial proximity doesn't always imply

functional relevance, detection of various acylcarnitines (molecules tentatively annotated as palmitoylcarnitine, oleoylcarnitine, linoleyl carnitine, and stearoyl carnitine) by MALDI-MSI, suggests that FFAs localized in the STB layer may be shuttled toward energy production.

v. The pathways that the authors reference in Fig 5 are not cited. For example, in Fig 5b, the authors highlight few reactions from the “estrogen synthesis pathway”, which resource was used (e.g. KEGG, Reactome) or if it’s curated from literature, where are the corresponding citations. If it is retrieved from a pathway databases, these reactions has to be viewed as part of the larger pathway entry to provide a better contextual view of the regulatory interactions involved. Checkout out PathView (<https://pathview.uncc.edu/>) for examples.

We thank the reviewer for their valuable insights. We revised our manuscript to include **Supplementary Figure 5** with reference pathways from KEGG and HumanCyc databases used for multiomics data reconstruction. All pathways discussed in our manuscript are accompanied by citations referencing previous studies that reported on tissue-specific pathways.

vi. In the discussion, the authors state that “integration multi-omics data can enhance confidence in lipid and metabolite identifications by colocalizing the detection of corresponding synthetic enzymes”. This is an oversimplification of metabolite/lipid identification because, as discussed before, co-detection can suggest functional relationship, but doesn’t guarantee one. The presence of a metabolite and its synthetic enzyme doesn’t confirm that the enzyme is actively synthesizing that metabolite at that location, because of many reasons. One being that some enzymes could generate structurally similar molecules as they can be non-specific. Other reasons include : post-translation regulation (expression vs activity) and transport

We agree with the reviewer and have revised our manuscript to address the reviewer’s comment.

For example, the following sentence was added to the discussion: Although co-detection does not imply co-regulation nor distinguish between pathway crosstalk, it provides spatial context that can be used to propose potential pathways, aiding in understanding basic biology and disease processes in specific tissue types.

vii. While the suggested modifications seem to improve the MIPI workflow experimentally, the data analysis performed seems rather primitive in comparison. A detailed characterization of metabolic and proteomics profile in selected sROIs are missing, also the spatial localization of the 14 selected ROIs on the adjacent sections are not reported. Other missing questions include:

1. How different are the STB/core regions on the same sections (intrareplicate variability) and in adjacent sections?

We have added these details to strength our manuscript. The mixed effects model directly quantifies and tests for differences in this quantity between tissue regions. Because samples cannot be considered independent, as there are repeated measures within each section, the mixed effects models appropriately partition out variability which can be attributed to within and between section variability and evaluate the difference in mean intensity or mean probability of detection after accounting for these sources of variability. We have added an analysis of inter- vs intra- section variability in the “Variability Partitioning: Inter- and Intra-Section Variability” section of the **Supplementary Statistical Methods**, to explicitly

calculate the percent variability in the data, for each protein, that can be attributed to intra-section variability.

“Variability Partitioning: Inter- and Intra-Section Variability” section of **Supplementary Statistical Methods** as it appears in the revised manuscript:

The intra-section variability and inter-section variability were considered in terms of two outcomes: peptide abundance and peptide detection. For each peptide, two models were fit: 1) a linear mixed effect model with abundance as the response variable with a fixed effect for region type and a random effect for tissue section, and 2) a generalized linear mixed effects model with peptide detection as the response variable was fit to the data with a conditional binomial distribution with a fixed effect for region type and a random effect for tissue section.

*For each peptide and model, the variability of measurements, after accounting for region type, was quantified using the partR2 package (Stoffel, Nakagawa, and Schielzeth 2021). The percent variability explained (R^2) can be attributed to the within section variability and the between section variability (while accounting for region type); lower R^2 means more consistent measurements. **Figure 7** shows the distribution of R^2 for within tissue measurements - R^2 for between tissue measurements, for the abundance and detection models, where negative values indicate more consistent measurements within a tissue. A majority of peptides show consistency in measurements within a tissue, while accounting for region type.*

Figure 7: Distribution of R^2 for within tissue measurements - R^2 for between tissue measurements, for the abundance and detection models, where negative values indicate more consistent measurements within a tissue.

2. How different are corresponding STB-Core pairs in each sections? For example Placenta_19_R1_core and Placenta_19_R1_STB vs Placenta_19_R2_core and Placenta_19_R2_STB.

We have added a “Sample to Sample Correlation” section in **Supplementary Statistical Methods**, where we calculated pairwise correlations between all samples.

“Sample to Sample Correlation” section of the **Supplementary Statistical Methods** as it appears in the revised manuscript:

Pearson’s correlation of sample abundance profiles were calculated. **Figure 8** gives a correlation heatmap with samples ordered by tissue, replicate pair, and subregion. There is very little clustering of samples by replicate pair. **Figure 9** gives the correlation heatmap with samples ordered by subregion, tissue, and replicate. From this plot, it is clear that the primary source of sample similarity is the subregion type (STB or Core).

Figure 8: Correlation heatmap of sample-to-sample correlation, ordered by tissue, replicate, and subregion.

Figure 9: Correlation heatmap of sample-to-sample correlation, ordered by subregion, tissue, and replicate.

3. How are the features correlated across the regions and sections. Metabolite-metabolite or protein-protein correlations? AND 4. Are the profiles observed in STB-Core regions preserved across adjacent sections?

We have provided the below details on how molecular features change across regions and sections.

Metabolite-metabolite correlations:

For metabolite-metabolite correlation we provided maximum signal intensity after root mean square normalization for the metabolites used for pathway integration. CVs of detected metabolites are in acceptable range of MALDI-MSI quantitation accuracy (~20%).

m/z	Name	Section 3	Section 4	CV (%)
		Maximum intensity (Peak Area)	Maximum intensity (Peak Area)	
400.3421	Palmitoylcarnitine	136657.375	171641.516	16
447.1278	Acetoacetic acid	163490.313	154546.891	4
449.1434	Hydroxybutyric acid	360541.094	268715.313	21
659.3207	Progesterone	22005.9297	25128.4219	9

Protein-protein correlations:

For protein-protein correlations our new of sample-to-sample correlations heatmaps (Fig. 9 in Supplementary Statistical Methods) illustrate profiles observed in STB and Core regions are preserved across adjacent sections since our primary source of sample similarity is the subregion type (STB or Core). In addition, all proteins discussed in the manuscript had consistent trends in STB-Core regions across adjacent sections due to the requirement of an adjusted p-value < 0.05.

Correlation heatmap of sample-to-sample correlation, ordered by subregion, tissue, and replicate.

Additional integrative analysis could also be performed given the provided data that could use relative intensity of the molecules in addition to just using co-detection.

We agree with the reviewer; however, the additional integrative interpretation is beyond the scope of our manuscript and would not fundamentally alter the intended deliverable.

viii. The paper and workflow in Figure 2 claims multi-omics data integration performed. This reads like an overstatement, as there is no integration performed. The authors have performed analysis of each omics dataset separately, and then manually picked metabolites and enzymes from the same pathways. However, normally data integration refers to an automated procedure.

We revised our Fig. 2 to change multi-omics data integration to pathway reconstruction.

Revised Fig. 2 as it appears in our manuscript:

*** Minor Comments ***

i. In line 68, the sentence “investigating morphological and molecular changes in the placenta could transform our discoveries in personalized therapies and novel therapeutics in maternal-fetal medicine” is a generic statement. Try to be more specific or use fewer generic statements.

We agree and have revised our manuscript accordingly.

Revision as it appears in the manuscript: Therefore, investigating both morphological and molecular changes can enhance our understanding of the processes and mechanisms during pregnancy that contribute to the health of both the mother and the fetus.

ii. In lines 69-72, the authors state in the introduction that only bulk samples were used in different omics studies. However, there are other spatial and/or single cell omics studies on the placenta that should be acknowledged as well. Please consider the following references:

1. <https://www.nature.com/articles/s41586-023-05869-0>

2. <https://www.nature.com/articles/s41588-023-01647-w>

We thank the reviewer for these suggestions and have added the references to our introduction.

Revision as it appears in the manuscript: Single-nucleus multi-omic profiling of human placental STB provided researchers insights into the heterogeneity of STB, allowing them to identify cellular trajectories during pregnancy[26]. Additionally, spatially resolved single-cell multiomic characterization allowed scientist to describe the complete trophoblast invasion trajectory in early pregnancy[27].

iii. In Lines 77-79, the authors state that “examining cellular diversity and tissue heterogeneity without dissociating the cells and disrupting their direct interactions can be done by mass spectrometry imaging (MSI)”. This applies to most if not all spatial omics technologies and is not exclusive to MSI. Please adjust accordingly.

We thank the reviewer for this comment. We have revised our manuscript accordingly.

iv. In line 110, use either omics or modalities – this line was edited.

v. Add citation of previously published MIPI workflow in lines 120-123. – we added the citation of previously published MIPI workflow.

vi. Providing data as a pdf table makes it harder to reproduce the analysis. Please provide it as a spreadsheet so that it would be easier to reanalyze. – We are happy to provide additional supplementary information (SI) as an excel spreadsheet. All files are automatically converted to PDF during the submission process, but uploaded as excel files.

vii. In Fig 3a, it’s hard to see the distinction between STB and Core, would it help if another image at better resolution and higher brightness is provided? – We change the brightness of the image in Figure 3a.

viii. In lines 169-171, the authors mention differential expression patterns of specific lipid classes. Where is the corresponding figure/table of such classes?. And which of the measured lipids are annotated to those classes?

We revised our manuscript to include Supplementary Table 3 with lipids annotated to diacylglycerophosphates, diacylglycerophosphoethanolamines, and diacylglycerophosphocholines and their spatial localization.

ix. In line 172, the authors mention that “stem villus region was enriched with lipid molecules that colocalized mainly with the villous core”. How was the enrichment calculated?. If no enrichment analysis was performed, please replace enriched with X lipid molecules were detected in high abundance (intensity > X) in X region compared with Y region. Also, how was colocalization calculated ?. If the authors use the colocalized molecules reported by METASPACE, please specify.

We didn't perform enrichment analysis; hence we revised our manuscript. Metaspace was used to compare the spatial distribution of ion images and visual inspection of colocalization.

x. In line 205-206, the authors mentioned that proteome profiling of 14 villous regions were performed informed by previous metabolomic imaging. I don't see any reference to those 14 regions in a figure or a table and their spatial localization on the respective tissues. Can you please add a reference to those regions?. And if they were metabolome informed, why are they not discussed in the previous results sections concerning metabolomics results?

Fourteen villous regions were collected for proteomics from 3 placental tissue sections (sections 19, 23, and 26), guided by metabolomic imaging from placental tissue sections in close proximity (sections 18 and 22). As we noted in our manuscript, the regions were excised from the adjacent sections (within 96 microns). We revised our manuscript to make it clear that it was metabolome-guided collection. We also added Supplementary Figure 3 for clarity. This figure shows the spatial localization of the regions mapped on metabolomic sections 18 and 22 for subsequent collection from sections 19, 23, and 26.

xi. In line 209, those proteins are differential expressed, not enriched based on the ANOVA test. Please rephrase accordingly.

*We rephrased the sentence “A total of 439 and 185 proteins were found to be differentially expressed in the STB and core sROIs, respectively (ANOVA-adjusted p-values < 0.05, **Supplemental Table 9 and Fig. 4b**)”*

xii. Some figures are cited as (Fig X) as in line 207 and others as (Figs. X) as in line 211. Please use consistent in-text citations.

Thank you for pointing out this discrepancy, now the main text uses “Figure X” and within paratheses (Fig. X)

Reviewer #4 (Remarks to the Author):

This paper by Veličković et al demonstrates an impressive spatial metabolomic and proteomic workflow demonstrated on placental tissue. For a spatial approach they obtain high level of sensitivity for lipids and metabolites measured. The examples where the approach was able to identify substrate enzyme and product in the same region provides confidence in the results and demonstrate their utility. The workflow involves technical advances to increase sensitivity e.g. on-tissue chemical derivatization.

We are grateful to the reviewer for taking time to review our article, and for their overall positive assessment. We addressed each comment as detailed below.

The paper uses the term “tentatively analyzed as” due to the technical limitations in annotating compounds. While caution is the correct approach here, this is nevertheless a limitation. How might a higher degree of certainty in the molecular annotation be achieved through technical innovation or combining data from other sources.

We agree with the reviewer that despite using a 12T FTICR mass spectrometer with an ultra-high-mass resolving power and mass accuracy, confident molecular annotation remains a challenge in this stand-alone method. Bulk data is often used to complement MALDI-MSI data, even though it lacks spatial information, the molecular signals are usually diluted, and molecular profiling is not completely comprehensive due to the usage of different ionization technique (electrospray ionization). Alternatively, we believe that confident molecular identification with a full spatial context of the tissue can be achieved by combining MALDI-MSI with trapped ion mobility spectrometry (TIMS). As such, tentatively annotated molecules from untargeted MALDI-MSI could be confirmed with targeted imaging approach by MALDI-TIMS-MS, along with a standards confirmation. We'll continue evolving our MIPI in that direction.

The distinction between trophoblast and core might not be as ambitious as it could be especially as regards the core which contains the greatest cellular heterogeneity.

Opposed to histologically guided proteome profiling, our MIPI approach focused on lipidome- and metabolome-guided imaging for complementary proteome profiling. In our study, we focused on identifying specific metabolic regions rather than individual cells. Molecules associated with ketone body metabolism demonstrated similar abundance throughout the entire mapped villous core compartment. Consequently, we did not further segment the core for subsequent proteome profiling.

In figure 3 there are structures that appear to be vessels from the histology and these could be easily identified as distinct regions?

We thank the reviewer for the opportunity to provide additional details regarding some of our metabolite and lipid images of interest. The MIPI approach described in our manuscript goes beyond histologically guided identification of regions of interest (ROIs) by utilizing spatial metabolite data to inform the subsequent selection and laser capture microdissection (LCM) of discrete placental functional units. Fetal blood vessels were not identified as distinct ROIs because they showed the same molecular profiles related to ketone body processing as the stromal cells. To elucidate this finding, we provided the below image where we overlaid ion and optical images and reduced ion image opacity so the molecular findings could be correlated with underneath histology. As it can be seen from the below image, for proteomic analysis we collected the pixels with the highest intensity related to ketone body processing from the mapped area, even though that area contained different cell types.

Metabolic imaging:

Acetoacetate; $[C_4H_6O_3 + C_{18}H_{22}N_2Br]^+$; 447.1278 m/z

Regarding lipidomic imaging, fetal blood vessels shared the spatial distribution of the molecules detected in stroma, as shown on the image below:

Lipidomic Imaging:

For this specific application, we used 25 μm and 15 μm step sizes for metabolomic and lipidomic imaging. We envision that MIPI will continue to evolve to explore biomolecular signatures in tissues at even higher resolutions, eventually to the cellular level. We acknowledge that higher spatial resolution should resolve features with finer details, allowing us to identify and characterize distinct areas more precisely.

The histology images, at least in the downloaded file, do not appear to be of high quality and I wonder if these could be improved? Details of the microscopy are not reported but the use of a slide scanner would produce high resolution tiled image of the whole structure. Also, could the current images be improved by adjusting the brightness?

We thank the reviewer for bringing these issues to our attention. Placental tissue sections were scanned on the LCM scanner using 20x-objective. We revised our figures to include the high-resolution histology images.

In figure 3 the expanded region could be shown with and without the overly so the underlying histology could be better understood, or at least a box placed on the histology image to show where it comes from.

We revised our Fig 3 to specify the spatial localization of the expanded region. We also provided an optical image without the overly, for the expanded region.

Revised figure:

This study represents a significant increase in resolution compared to previous work, but the holy grail would be a true cellular analysis (for instance separating syncytiotrophoblast and cytotrophoblast). The discussion could have included a more comprehensive limitations section addressing current limitations (including confidence in molecular annotation) and outlining those limiting future progress such as further increases in resolution.

We agree with your observation and have made necessary modifications throughout the manuscript to emphasize our awareness of these limitations. We believe that our revised limitations paragraph in the discussion adds valuable context to our work and future areas of improvement.

Could the authors comment further on the potential for this approach to be used to compare healthy and pathological tissue?

Yes, we are excited to utilize the MIPI approach in future studies to gain molecular insights into both healthy and pathological placenta tissues. It has been reported that placental metabolic alterations are commonly observed in a variety of pregnancy complications and diseases. For instance, mitochondrial fatty acid oxidation disorder in a fetus is associated with maternal diseases of pregnancy. Additionally, altered steroid metabolism can lead to several pregnancy complications and pregnancy loss. Therefore, in our placenta study we were particularly interested in capturing and spatially resolving pathways related to steroid and fatty acid metabolism. With this proof-of-concept, we believe that the MIPI approach can be used to capture molecular dynamics and distribution across the tissue, deciphering biological pathways and their pathologic alterations across different gestational stages and maternal conditions.

As a minor points:

Line 287, rather than failure to colocalize could in this region a higher flux from substrate to product could result resulting in levels of substrate lower than the level of detection.

Since the mentioned line does not discuss substrate/product conversion, but spatial localization of free fatty acid (FFA) and their corresponding acylcarnitines, we hope we adequately addressed the reviewer's comment below.

Detection of oleoylcarnitine and linoleylcarnitine in STB and the corresponding derivatized fatty acids across the entire villous functional unit supports the concept of FFAs transfer across the placenta. We also detected docosahexaenoic acid (DHA), an essential FFA that showed different spatial pattern than mentioned fatty acids. For review only, we've included ion image of essential fatty acid DHA that was detected in the STB region only. Previous studies* have reported that only a small proportion of certain FFA are directly transferred across the placenta whereas DHA showed reduced mobilization in placentae of obese women, resulting in lower DHA levels in the fetal capillaries, which is in line with our imaging data from placentae of obese mother that capture DHA in STB subregion only. Our advanced workflow enabled us to boost the sensitivity and detect FFAs in their derivatized form, capturing endogenous FFAs. Therefore, we believe that we accurately visualized the placental tissue snapshot in time, and that we didn't approach a limit of detection, although it's always possible.

* Hirschmugl, B., et al., Placental mobilization of free fatty acids contributes to altered materno-fetal transfer in obesity. *International Journal of Obesity*, 2021. 45(5): p. 1114-1123.

The syncytiotrophoblast is a syncytium and I would not refer to STB cells.

We thank the reviewer for this important comment. We revised our manuscript to make it clear that the syncytiotrophoblast (STB) is a syncytium.

The figure 1 legend could be reworded slightly to make clear that the syncytiotrophoblast is not a layer of cells but a syncytium.

We reworded the Fig. 1 legend to make clear that the syncytiotrophoblast is not a layer of cells but a syncytium.

NCOMMS-24-50998A

We thank all reviewers for taking the time to review our revised article and provide their feedback. We have revised **Figure 5c** in response to reviewer #4's comment; however, the rest of the manuscript remains unchanged. See remarks to the reviewer #1 below.

Reviewer #1 (Remarks to the Author):

Although the authors took a lot of effort to answer the many comments of all reviewers, I still do not see this paper as Nature Communication material since:

We thank the reviewer for their feedback; however, we have a different perspective on this specific point. In our responses below, we highlight the novelties in the technical approach, biological integration, and data interpretation. Collectively, we believe these elements make our article well-suited for Nature Communications.

* The main workflow part is not new at all and having a higher amount of proteins from extraction is not enough for a high impact paper. The number of tissues should also increase to state their conclusions confidently.

While improved spatial resolution and higher proteome coverage were significant advancements of our MIPI approach, they are not the only novelty of our manuscript. Implementing on-tissue chemical derivatization (OTCD) in our metabolomic imaging of human placenta tissue significantly expanded our metabolome coverage. We were able to capture highly inert steroids and fatty acids in their endogenous form across the placenta sections, which is the first imaging of its kind. Additionally, complementary lipidomic imaging expanded detection coverage, offering more comprehensive data. Further spatial proteome profiling with relatively high proteome coverage allowed us to capture corresponding enzymes. Integrating data from multiple omics levels allowed us to roll up from molecular-level to the pathway-level, enhancing our understanding of the compartmentalization of placental villous functional units by revealing biological processes and metabolic pathways associated with STB and core (i.e., the reconstruction of the estrogen synthesis and signaling pathway, fatty acid transport and energy production pathway, and *de novo* ceramide synthesis pathway in the villous STB compartment, and a proposed new ketone body oxidation pathway in the villous core compartment). Through this study, we demonstrated the capability of integrated MIPI data to unveil biological novelties by providing a comprehensive view of underlying pathway activities, highlighting the associated differences between villous STB and core compartments at a micrometer-scale level.

* Although a very high mass resolution MS is used, there are still tentative ID's which raises questions. MS/MS analysis could have been done.

MS/MS obtained from bulk data is often used to complement MALDI-MSI data. However, it lacks spatial information, the molecular signals are usually diluted, and molecular profiling is not fully comprehensive

due to using a different ionization technique (electrospray ionization). Alternatively, integrating multi-omics data can provide a higher degree of certainty in the molecular annotation of lipids and metabolites by colocalizing the detection of their corresponding enzymes. Therefore, in our work, we mapped enzymes and their corresponding substrates and products across multiple sections to suggest functional relationships, providing pathway-level insights at micrometer-scale resolution.

* the spatial resolution used is not high enough to draw the conclusions they do from their data. This also includes the precision of the LCM and the H&E now added (not convincing at all to do LCM on such a small area).

We respectfully disagree with the reviewer's comment regarding the spatial resolution being insufficient for drawing our conclusions. Our study examined placental functional units with a focus on metabolic regions, not individual cells. In our article, we demonstrated that in situ metabolomic imaging at a 25 μm spatial resolution is adequate for spatially resolving molecules and characterizing the inner villous (core) and outer villous (STB) compartments within placental villous functional units. This is further substantiated by our additional lipidomic imaging, conducted at a higher spatial resolution of 15 μm , followed by molecular segmentation that identified molecular differences of those two distinct villous compartments (STB and core), even though placental villous functional units were imaged with the higher spatial resolution.

All post-MALDI sections were H&E-stained and overlaid with MALDI-MSI ion images to correlate morphology and spatial localization of the resolved molecules. For that matter, we included H&E images in our original submission, they were not added in our revised manuscript.

We appreciate the reviewer's recognition of our meticulous work to precisely map and excise the micrometer-scale subregions of identified villous functional units across the depth of adjacent sections. By leveraging the seamless integration of microPOTS with LCM, we microdissected and collected a 10,000 μm^2 tissue area. This is a relatively large area, considering that we've previously demonstrated our ability to microdissect tissue pieces with square lateral dimensions as small as 20 μm (400 μm^2)*. Although microdissection of tissue as small as 20 μm in diameter (single-cell scale) is feasible, in practice much larger samples are still required to yield greater proteome coverage.

**Ying Zhu et al., Spatially Resolved Proteome Mapping of Laser Capture Microdissected Tissue with Automated Sample Transfer to Nanodroplets, Molecular & Cellular Proteomics, Volume 17, Issue 9, 1864 – 1874.*

Reviewer #2 (Remarks to the Author):

The authors have performed an extensive revision of the text and figures which substantially improved the manuscript.

I believe that my concerns were substantially addressed.

We appreciate the reviewer's constructive comments that improved our manuscript.

Reviewer #3 (Remarks to the Author):

Reviewer #4 (Remarks to the Author):

The manuscript has been clarified and improved in response to the referees comments. The points I raised have been largely addressed. I do not have any further substantial comments.

some inset higher refs images might be nice in figure 5c

Thank you for this suggestion. We have revised **Figure 5c** to include expanded regions, which reveal the underlying morphology in greater detail and higher resolution.